# Tensor Completion Made Practical

**Allen Liu**
Massachusetts Institute of Technology
Cambridge, MA 02139
cliu568@mit.edu

**Ankur Moitra**\*
Department of Mathematics, Massachusetts Institute of Technology
Cambridge, MA 02139
moitra@mit.edu

## Abstract

Tensor completion is a natural higher-order generalization of matrix completion where the goal is to recover a low-rank tensor from sparse observations of its entries. Existing algorithms are either heuristic without provable guarantees, based on solving large semidefinite programs which are impractical to run, or make strong assumptions such as requiring the factors to be nearly orthogonal. In this paper we introduce a new variant of alternating minimization, which in turn is inspired by understanding how the progress measures that guide convergence of alternating minimization in the matrix setting need to be adapted to the tensor setting. We show strong provable guarantees, including showing that our algorithm converges linearly to the true tensors even when the factors are highly correlated and can be implemented in nearly linear time. Moreover our algorithm is also highly practical and we show that we can complete third order tensors with a thousand dimensions from observing a tiny fraction of its entries. In contrast, and somewhat surprisingly, we show that the standard version of alternating minimization, without our new twist, can converge at a drastically slower rate in practice.

## 1  Introduction

In this paper we study the problem of recovering a low-rank tensor from sparse observations of its entries. In particular, suppose

$$T = \sum_{i=1}^{r} \sigma_i \, x_i \otimes y_i \otimes z_i$$

Here $\{x_i\}_i$, $\{y_i\}_i$ and $\{z_i\}_i$ are called the *factors* in the low rank decomposition and the $\sigma_i$'s are scalars, which allows us to assume without loss of generality that the factors are unit vectors. Additionally $T$ is called a *third order* tensor because it is naturally represented as a three-dimensional array of numbers. Now suppose each entry of $T$ is revealed independently with some probability $p$. Our goal is to accurately estimate the missing entries with $p$ as small as possible.

Tensors completion is a natural generalization of the classic matrix completion problem [7]. Similarly, it has a wide range of applications including in recommendation systems [33], signal and image processing [20, 19, 24, 4], data analysis in engineering and the sciences [26, 18, 29, 31] and harmonic

analysis [27]. However, unlike matrix completion, there is currently a large divide between theory and practice. Algorithms with rigorous guarantees either rely on solving very large semidefinite programs [3, 25], which is impractical, or else need to make strong assumptions that are not usually satisfied [6], such as assuming that the factors are nearly orthogonal, which is a substantial restriction on the model. In contrast, the most popular approach in practice is *alternating minimization* where we fix two out of three sets of factors and optimize over the other

$$(\widehat{z}_1, \ldots, \widehat{z}_r) = \arg\min_{z_1, \ldots, z_r} \left\| \left( T - \sum_{i=1}^{r} \widehat{x}_i \otimes \widehat{y}_i \otimes z_i \right) \Big|_S \right\|_2^2 \tag{1}$$

Here $S$ is the set of observations and we use $X\Big|_S$ to denote restricting a tensor $X$ to the set of entries in $S$. We then update our estimates, and optimize over a different set of factors, continuing in this fashion until convergence. A key feature of alternating minimization that makes it so appealing in practice is that it only needs to store $3r$ vectors along with the observations, and never explicitly writes down the entire tensor. Unfortunately, not much is rigorously known about alternating minimization for tensor completion, unlike for its matrix counterpart [16, 14].

In this paper we introduce a new variant of alternating minimization for which we can prove strong theoretical guarantees. Moreover we show that our algorithm is highly practical. We can complete third order tensors with a thousand dimensions from observing a tiny fraction of its entries. We observe experimentally that, in many natural settings, our algorithm takes an order of magnitude fewer iterations to converge than the standard version of alternating minimization.

## 1.1 Prior Results

In matrix completion, the first algorithms were based on finding a completion that minimizes the nuclear norm [7]. This is in some sense the best convex relaxation to the rank [8], and originates from ideas in compressed sensing [10]. There is a generalization of the nuclear norm to the tensor setting. Thus a natural approach [32] to completing tensors is to solve the convex program

$$\min \|\widehat{T}\|_* \text{ s.t. } \widehat{T}\Big|_S = T\Big|_S$$

where $\|\cdot\|_*$ is the tensor nuclear norm. Unfortunately the tensor nuclear norm is hard to compute [13, 15], so this approach does not lead to any algorithmic guarantees.

Barak and Moitra [3] used a semidefinite relaxation to the tensor nuclear norm and showed that an $n \times n \times n$ incoherent tensor of rank $r$ can be recovered approximately from roughly $rn^{3/2}$ observations. Moreover they gave evidence that this bound is tight by showing lower bounds against powerful families of semidefinite programming relaxations as well as relating the problem of completing approximately low-rank tensors from few observations to the problem of refuting a random constraint satisfaction problem with few clauses [9]. This is a significant difference from matrix completion in the sense that here there are believed to be fundamental computational vs. statistical tradeoffs whereby any efficient algorithm must use a number of observations that is larger by a polynomial factor than what is possible information-theoretically. Their results, however, left open two important questions:

**Question 1.** *Are there algorithms that achieve exact completion, rather than merely getting most of the missing entries mostly correct?*

**Question 2.** *Are there much faster algorithms that still have provable guarantees – ideally ones that can actually be implemented in practice?*

For the first question, Potechin and Steurer [25] gave a refined analysis of the semidefinite programming approach through which they gave an exact completion algorithm when the factors in the low rank decomposition are orthogonal. Jain and Oh [17] obtain similar guarantees via an alternating minimization-based approach. For matrices, orthogonality of the factors can be assumed without loss of generality. But for tensors, it is a substantial restriction. Indeed, one of the primary applications of tensor decomposition is to parameter learning where the fact that the decomposition is unique even when the factors can be highly correlated is essential [2]. Xia and Yuan [30] gave an algorithm based on optimization over the Grassmannian and they claimed it achieves exact completion in polynomial time under mild conditions. However no bound on the number of iterations was given, and it is only

known that each step can be implemented in polynomial time. For the second question, Cai et al. [6] gave an algorithm based on nonconvex optimization that runs in nearly linear time up to a polynomial in $r$ factor. (In the rest of the paper we will think of $r$ as constant or polylogarithmic and thus we will omit the phrase "up to a polynomial in $r$ factor" when discussing the running time.) Moreover their algorithm achieves exact completion under the somewhat weaker condition that the factors are nearly orthogonal. Notably, their algorithm also works in the noisy setting where they showed it nearly achieves the minimax optimal prediction error for the missing entries.

There are also a wide variety of heuristics. For example, there are many other relaxations for the tensor nuclear norm based on flattening the tensor into a matrix in different ways [11]. However we are not aware of any rigorous guarantees for such methods that do much better than completing each slice of the tensor as its own separate matrix completion problem. Such methods require a number of observations that is a polynomial factor larger than what is needed by other algorithms.

## 1.2 Our Results

In this paper we introduce a new variant of alternating minimization that is not only highly practical but also allows us to resolve many of the outstanding theoretical problems in the area. Our algorithm is based on some of the key progress measures behind the theoretical analysis of alternating minimization in the matrix completion setting. In particular, Jain et al. [16] and Hardt [14] track the principal angle between the true subspace spanned by the columns of the unknown matrix and that of the estimate. They prove that this distance measure decreases geometrically. In Hardt's [14] analysis this is based on relating the steps of alternating minimization to a noisy power method, where the noise comes from the fact that we only partially observe the matrix that we want to recover.

We observe two simple facts. First, if we have bounds on the principal angles between the subspaces spanned by $x_1, \ldots, x_r$ and $\widehat{x}_1, \ldots, \widehat{x}_r$ as well as between the subspaces spanned by $y_1, \ldots, y_r$ and $\widehat{y}_1, \ldots, \widehat{y}_r$ then it does *not* mean we can bound the principal angle between the subspaces spanned by

$$x_1 \otimes y_1, \ldots, x_r \otimes y_r \text{ and } \widehat{x}_1 \otimes \widehat{y}_1, \ldots, \widehat{x}_r \otimes \widehat{y}_r$$

However if we take all pairs of tensor products, and instead consider the principal angle between the subspaces spanned by

$$\{x_i \otimes y_j\}_{i,j} \text{ and } \{\widehat{x}_i \otimes \widehat{y}_j\}_{i,j}$$

then we can bound the principal angle (see Observation 3.1). This leads to our new variant of alternating minimization, which we call KRONECKER ALTERNATING MINIMIZATION, where the new update rule is:

$$\{\widehat{z}_{i,j}\} = \arg\min_{z_{i,j}} \left\| \left( T - \sum_{1 \leq i,j \leq r} \widehat{x}_i \otimes \widehat{y}_j \otimes z_{i,j} \right) \Big|_S \right\|_2^2 \tag{2}$$

In particular, we are solving a least squares problem over the variables $z_{i,j}$ by taking the Kronecker product of $\widehat{X}$ and $\widehat{Y}$ where $\widehat{X}$ is a matrix whose columns are the $\widehat{x}_i$'s and similarly for $\widehat{Y}$. In contrast the standard version of alternating minimization takes the Khatri-Rao product[2].

This modification increases the number of rank one terms in the decomposition from $r$ to $r^2$. We show that we can reduce back to $r$ without incurring too much error by finding the best rank $r$ approximation to the $n \times r^2$ matrix of the $\widehat{z}_{i,j}$'s. We combine these ideas with methods for initializing alternating minimization, building on the work of Montanari and Sun [22], along with a post-processing algorithm to solve the non-convex exact completion problem when we are already very close to the true solution. Our main result is:

**Theorem 1.1** (Informal version of Theorem 3.2). *Suppose $T$ is an $n \times n \times n$ low-rank, incoherent, well-conditioned tensor and its factors are robustly linearly independent. There is an algorithm that runs in nearly linear time in the number of observations and exactly completes $T$ provided that each entry is observed independently with probability $p$ where*

$$p \geq C\frac{r^{O(1)}}{n^{3/2}}$$

*Moreover the constant $C$ depends polynomially on the incoherence, the condition number and the inverse of the lower bound on how far the factors are from being linearly dependent.*

We state the assumptions precisely in Section 2.1. This algorithm combines the best of many worlds:

(1) It achieves **exact completion**, even when the factors are highly correlated.

(2) It runs in **nearly linear time** in terms of the number of observations.

(3) The alternating minimization phase **converges at a linear rate**.

(4) It **scales to thousands of dimensions**, whereas previous experiments had been limited to about a hundred dimensions.

(5) Experimentally, in the presence of noise, it still achieves strong guarantees. In particular, it achieves **nearly optimal prediction error**.

We believe that our work takes an important and significant step forward to making tensor completion practical, while nevertheless maintaining strong provable guarantees.

## 2  Preliminaries

### 2.1  Model and Assumptions

As usual in matrix and tensor completion, we will need to make an incoherence assumption as otherwise the tensor could be mostly zero and have a few large entries that we never observe. We will also assume that the components of the tensor are not too close to being linearly dependent as otherwise the tensor will be degenerate. (Even if we fully observed it, it would not be clear how to decompose it.)

**Definition 2.1.** *Given a subspace $V \subset \mathbb{R}^n$ of dimension $r$, we say $V$ is $\mu$-incoherent if the projection of any standard basis vector $e_i$ onto $V$ has length at most $\sqrt{\mu r/n}$*

**Assumptions 1.** *Consider an $n \times n \times n$ tensor with a rank $r$ CP decomposition*

$$T = \sum_{i=1}^{r} \sigma_i (x_i \otimes y_i \otimes z_i)$$

*where $x_i, y_i, z_i$ are unit vectors and $\sigma_1 \geq \cdots \geq \sigma_r > 0$. We make the following assumptions*

- **Robust Linear Independence**: *The smallest singular value of the matrix with columns given by $x_1, \ldots, x_r$ is at least $c$. The same is true for $y_1, \ldots, y_r$ and $z_1, \ldots, z_r$.*

- **Incoherence**: *The subspace spanned by $x_1, \ldots, x_r$ is $\mu$-incoherent. The same is true for $y_1, \ldots, y_r$ and $z_1, \ldots, z_r$.*

*Finally we observe each entry independently with probability $p$ and our goal is to recover the original tensor $T$.*

We will assume that for some sufficiently small constant $\delta$,

$$\max\left(r, \frac{\sigma_1}{\sigma_r}, \frac{1}{c}, \mu\right) \leq n^{\delta}.$$

In other words, we are primarily interested in how the number of observations scales with $n$, provided that it has polynomial dependence on the other parameters.

## 3  Our Algorithms

Here we will give our full algorithm for tensor completion, which consists of the three phases, initialization through spectral methods, our new variant of alternating minimization, and finally a post-processing step to recover the entries of the tensor exactly (when there is no noise). We defer a formal description of the initialization and post-processing steps to the Section A of the appendix. In

---

**Algorithm 1** FULL EXACT TENSOR COMPLETION

---

**Input:** Let $\widehat{T}$ be a sample where we observe each entry of $T$ independently with probability $p$

Let $p_1, p_2, p_3$ be parameters with $p_1 + p_2 + p_3 \leq p$

Split $\widehat{T}$ into three samples $\widehat{T_1}, \widehat{T_2}, \widehat{T_3}$ using Claim C.7 such that for $i = 1, 2, 3$, in $\widehat{T_i}$ each entry is observed with probability $p_i$

Run INITIALIZATION using $\widehat{T_1}$ to obtain initial estimates $V_x^0, V_y^0, V_z^0$ for the subspaces $V_x, V_y, V_z$

Run KRONECKER ALTERNATING MINIMIZATION using $\widehat{T_2}$ and initial subspace estimates $V_x^0, V_y^0, V_z^0$ to obtain refined subspace estimates $\widehat{V_x}, \widehat{V_y}, \widehat{V_z}$

Run POST-PROCESSING VIA CONVEX OPTIMIZATION using $\widehat{T_3}$ and the subspace estimates $\widehat{V_x}, \widehat{V_y}, \widehat{V_z}$ from the previous step.

---

Section H, we will show that our algorithm can be implemented so that it runs in nearly linear time in terms of the number of observations.

We will use the following notation in the detailed description of our alternating minimization algorithm. Let $U_x(T)$ denote the unfolding of $T$ into an $n \times n^2$ matrix where the dimension of length $n$ corresponds to the $x_i$. Define $U_y(T), U_z(T)$ similarly.

---

**Algorithm 2** KRONECKER ALTERNATING MINIMIZATION

---

**Input:** Let $\widehat{T}$ be a sample where we observe each entry of $T$ independently with probability $p_2$
**Input:** Let $V_x^0, V_y^0, V_z^0$ be initial subspace estimates that we are given

Set $k = 10^2 \log \frac{n\sigma_1}{c\sigma_r}$

Let $p'$ be a parameter such that $kp' \leq p_2$

Split $\widehat{T}$ into independent samples $\widehat{T_1}, \ldots \widehat{T_k}$ using Claim C.7 such that

- For $1 \leq i \leq k$, in $\widehat{T_i}$ each entry is revealed with probability $p'$

**for** $t = \{0, 1, \ldots, k-1\}$ **do**

  Let $B_t$ be an $r^2 \times n^2$ matrix whose rows are an orthonormal basis for $V_y^t \otimes V_z^t$

  Consider the sample $\widehat{T}_{t+1}$ and let $S_{t+1}$ be the set of observed entries

  Let $H_{t+1}$ be the solution to $\min_H \left\| \left( U_x(\widehat{T}_{t+1}) - HB_t \right) \big|_{S_{t+1}} \right\|_2^2$

  Let $V_x^{t+1}$ be the space spanned by the top $r$ left singular vectors of $H_{t+1}$
  Compute $V_y^{t+1}$ from $V_x^t, V_z^t$ and compute $V_z^{t+1}$ from $V_x^t, V_y^t$ similarly

**Output:** $V_x^k, V_y^k, V_z^k$

---

**Alternating Minimization, with a Twist**  Recall the standard formulation of alternating minimization in tensor completion, given in Equation 1. Unfortunately, this approach is difficult to analyze from a theoretical perspective. In fact, in Section 4 we observe experimentally that it can indeed get stuck. Moreover, even if we add randomness by looking at a random subset of the observations in each step, it converges at a prohibitively slow rate when the factors of the tensor are correlated. Instead, in Equation 2 we gave a subtle modification to the alternating minimization steps that prevents it from getting stuck. We then update $\widehat{z}_1, \ldots, \widehat{z}_r$ to be the top $r$ left singular vectors of the $n \times r^2$ matrix with columns given by the $\widehat{z}_{i,j}$. With this modification, we will be able to prove strong theoretical guarantees, even when the factors of the tensor are correlated.

The main tool in the analysis of our alternating minimization algorithm is the notion of the principal angles between subspaces. Intuitively, the principal angle between two $r$-dimensional subspaces $U, V \subset \mathbb{R}^n$ is the largest angle between some vector in $U$ and the subspace $V$. For matrix completion, Jain et al. [16] and Hardt [14] analyze alternating minimization by tracking the principal angles between the subspace spanned by the top $r$ singular vectors of the true matrix and of the estimate at each step. Our analysis follows a similar pattern. We rely on the following key observation (whose proof is in Section C) as the starting point for our work:

**Observation 3.1.** *Given subspaces $U, V \in \mathbb{R}^n$ of the same dimension, let $\alpha(U, V)$ be the sine of the principal angle between $U$ and $V$. Suppose we have subspaces $U_1, V_1 \subset \mathbb{R}^{n_1}$ of dimension $d_1$ and*

$U_2, V_2 \subset \mathbb{R}^{n_2}$ *of dimension $d_2$, then*

$$\alpha(U_1 \otimes U_2, V_1 \otimes V_2) \le \alpha(U_1, V_1) + \alpha(U_2, V_2).$$

Thus if the subspaces spanned by the estimates $(\widehat{x}_1, \ldots, \widehat{x}_r)$ and $(\widehat{y}_1, \ldots, \widehat{y}_r)$ are close to the subspaces spanned by the true vectors $(x_1, \ldots, x_r)$ and $(y_1, \ldots, y_r)$, then the solution for $\{z_{i,j}\}$ in Equation 2 will have small error – i.e. $\sum_{1 \le i,j \le r} \widehat{x}_i \otimes \widehat{y}_j \otimes z_{i,j}$ will be close to $T$. This means that the top $r$ principal components of the matrix with columns given by $\{\widehat{z}_{i,j}\}$ must indeed be close to the space spanned by $z_1, \ldots, z_r$. For more details, see Corollary E.7; this result allows us to prove that our alternating minimization steps make progress in reducing the principal angles of our subspace estimates.

On the other hand, Observation 3.1 does *not* hold if the tensor product is replaced with the Khatri-Rao product, which is what would be the natural route towards analyzing an algorithm that uses Equation 1. To see this consider

$$U_1 = V_1 = U_2 = \begin{bmatrix} 1 & 0 \\ 0 & 1 \\ 0 & 0 \end{bmatrix}, V_2 = \begin{bmatrix} 0 & 1 \\ 1 & 0 \\ 0 & 0 \end{bmatrix}$$

(where we really mean that $U_1, V_1, U_2, V_2$ are the subspaces spanned by the columns of the respective matrices). Then the principal angles between $U_1, V_1$ and and $U_2, V_2$ are zero yet the principal angle between $U_1 \otimes U_2$ and $V_1 \otimes V_2$ is $\frac{\pi}{2}$ i.e. they are orthogonal. This highlights another difficulty in analyzing Equation 1, namely that the iterates in the alternating minimization depend on the order of $(\widehat{x}_1, \ldots, \widehat{x}_r)$ and $(\widehat{y}_1, \ldots, \widehat{y}_r)$ and not just the subspaces themselves.

**Initialization and Cleanup** For the full theoretical analysis of our algorithm, in addition to the alternating minimization, we will need two additional steps. First, we obtain a sufficiently good initialization by building on the work of Montanari and Sun [22]. We use their algorithm for estimating the subspaces spanned by the $\{x_i\}, \{y_i\}$ and $\{z_i\}$. However we then slightly perturb those estimates to ensure that our initial subspaces are incoherent.

Next, in order to obtain exact completion rather than merely being able to estimate the entries to any desired inverse polynomial accuracy, we prove that exact tensor completion reduces to an optimization problem that is convex once we have subspace estimates that are sufficiently close to the truth (see Section A.1 for a further discussion on this technicality). To see this, applying robustness analyses of tensor decomposition [21] to our estimates at the end of the alternating minimization phase it follows that we can not only estimate the subspaces but also the entries and rank one components in the tensor decomposition to any inverse polynomial accuracy. Now if our estimates are given by $\{\widehat{\sigma}_i\}, \{\widehat{x}_i\}, \{\widehat{y}_i\}, \{\widehat{z}_i\}$ we can write the expression

$$\widehat{T}(\Delta) = \sum_{i=1}^{r} (\widehat{\sigma}_i + \Delta_{\sigma_i})(\widehat{x}_i + \Delta_{x_i}) \otimes (\widehat{y}_i + \Delta_{y_i}) \otimes (\widehat{z}_i + \Delta_{z_i}) \tag{3}$$

and attempt to solve for $\Delta_{\sigma_i}, \Delta_{x_i}, \Delta_{y_i}, \Delta_{z_i}$ that minimize

$$\left\| \left( \widehat{T}(\Delta) - T \right) \Big|_S \right\|_2^2.$$

The key observation is that since we can ensure $\{\widehat{\sigma}_i\}, \{\widehat{x}_i\}, \{\widehat{y}_i\}, \{\widehat{z}_i\}$ are all close to their true values, all of $\Delta_{\sigma_i}, \Delta_{x_i}, \Delta_{y_i}, \Delta_{z_i}$ are small and thus $\widehat{T}(\Delta)$ can be approximated well by its linear terms. If we only consider the linear terms, then solving for $\Delta_{\sigma_i}, \Delta_{x_i}, \Delta_{y_i}, \Delta_{z_i}$ is simply a linear least squares problem. Intuitively, $\left\| \left( \widehat{T}(\Delta) - T \right) \Big|_S \right\|_2^2$ must also be convex because the contributions from the non-linear terms in $\widehat{T}(\Delta)$ are small. The precise formulation that we use in our algorithm will be slightly different from Equation 3 because in Equation 3, there are certain redundancies i.e. ways to set $\Delta_{\sigma_i}, \Delta_{x_i}, \Delta_{y_i}, \Delta_{z_i}$ that result in the same $\widehat{T}(\Delta)$. See Section G and in particular, Lemma G.2 for details.

We will now state our main theorem:

**Theorem 3.2.** *The* FULL EXACT TENSOR COMPLETION *algorithm run with the following parameter settings:*

$$p = 2 \left( \frac{\mu r \log n}{c} \cdot \frac{\sigma_1}{\sigma_r} \right)^{300} \cdot \frac{1}{n^{3/2}}, \; p_1 = \left( \frac{2\mu r \log n}{c^2} \cdot \frac{\sigma_1^2}{\sigma_r^2} \right)^{10} \cdot \frac{1}{n^{3/2}}$$

$$p_2 = \left( \frac{\mu r \log n}{c} \cdot \frac{\sigma_1}{\sigma_r} \right)^{300} \cdot \frac{1}{n^{3/2}}, \; p_3 = 2 \frac{\log^2 n}{n^2} \left( \frac{10r\mu}{c} \cdot \frac{\sigma_1}{\sigma_r} \right)^{10}$$

*successfully outputs $T$ with probability at least $0.9$. Furthermore, the algorithm can be implemented to run in $n^{3/2}\mathrm{poly}(r, \log n, \sigma_1/\sigma_r, \mu, 1/c)$ time.*

## 4 Experiments

In this section, we describe our experimental results and in particular how our algorithm compares to existing algorithms. First, it is important to notice that how well an algorithm performs can sometimes depend a lot on properties of the low-rank tensor. In our experiments, we find that there are many algorithms that succeed when the factors are nearly orthogonal but degrade substantially when the factors are correlated. To this end, we ran experiments in two different setups:

- Uncorrelated tensors: generated by taking $T = \sum_{i=1}^4 x_i \otimes y_i \otimes z_i$ where $x_i, y_i, z_i$ are random unit vectors.
- Correlated tensors: generated by taking $T = \sum_{i=1}^4 0.5^{i-1} x_i \otimes y_i \otimes z_i$ where $x_1, y_1, z_1$ are random unit vectors and for $i > 1$, $x_i, y_i, z_i$ are random unit vectors that have covariance $\sim 0.88$ with $x_1, y_1, z_1$ respectively.

Unfortunately many algorithms in the literature either cannot be run on real data, such as ones based on solving large semidefinite programs [3, 25] and for others no existing code was available. Moreover some algorithms need to solve optimization problems with as many as $n^3$ constraints [30] and do not seem to be able to scale to the sizes of the problems we consider here. Instead, we primarily consider two algorithms: a variant of our algorithm, which we call KRONECKER COMPLETION, and STANDARD ALTERNATING MINIMIZATION. For KRONECKER COMPLETION, we randomly initialize $\{\widehat{x_i}\}, \{\widehat{y_i}\}, \{\widehat{z_i}\}$ and then run alternating minimization with updates given by Equation 2. For given subspace estimates, we execute the projection step of POST-PROCESSING VIA CONVEX OPTIMIZATION to estimate the true tensor (omitting the decomposition and convex optimization). It seems that neither the initialization nor the post-processing steps are needed in practice. For STANDARD ALTERNATING MINIMIZATION, we randomly initialize $\{\widehat{x_i}\}, \{\widehat{y_i}\}, \{\widehat{z_i}\}$ and then run alternating minimization with updates given by Equation 1. To estimate the true tensor, we simply take $\sum_{i=1}^r \widehat{x_i} \otimes \widehat{y_i} \otimes \widehat{z_i}$. The code for all of the experiments can be found in Section I.

For alternating minimization steps, we use a random subset consisting of half of the observations. We call this subsampling. Subsampling appears to improve the performance, particularly of STANDARD ALTERNATING MINIMIZATION. We discuss this in more detail below.

We ran KRONECKER COMPLETION and STANDARD ALTERNATING MINIMIZATION for $n = 200, r = 4$ and either $50000$ or $200000$ observations. We ran $100$ trials and took the median normalized MSE i.e. $\frac{\|T_{\mathrm{est}} - T\|_2}{\|T\|_2}$. The results for these experiments are in Figure 1. For both algorithms, the error converges to zero rapidly for uncorrelated tensors. However, for correlated tensors, the error for KRONECKER COMPLETION converges to zero at a substantially faster rate than the error for STANDARD ALTERNATING MINIMIZATION. Compared to STANDARD ALTERNATING MINIMIZATION, the runtime of each iteration of KRONECKER COMPLETION is larger by a factor of roughly $r$ (here $r = 4$). However, the error for KRONECKER COMPLETION converges to zero in around $30$ iterations while the error for STANDARD ALTERNATING MINIMIZATION fails to converge to zero after $1000$ iterations, despite running for nearly $10$ times as long. Naturally, we expect the convergence rate to be faster when we have more observations. Our algorithm exhibits this behavior. On the other hand, for STANDARD ALTERNATING MINIMIZATION, the convergence rate is actually slower with $200000$ observations than with $50000$.

In the rightmost plot, we run the same experiments with correlated tensors without subsampling. Note the large oscillations and drastically different behavior of STANDARD ALTERNATING MINIMIZATION with $50000$ observations.

**Error over time for** KRONECKER COMPLETION **and** STANDARD ALTERNATING MINIMIZATION

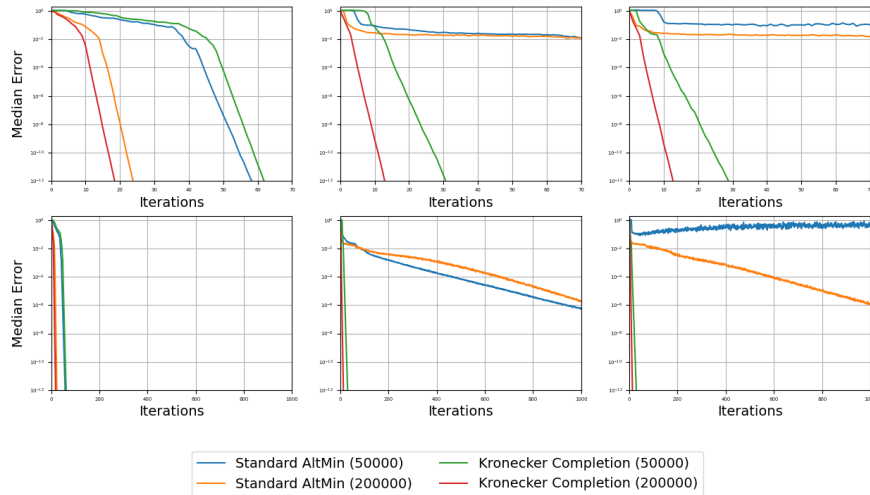

Figure 1: Top Row: The plot on the left is for uncorrelated tensors. The plot in the middle is for correlated tensors and with subsampling. The plot on the right is for correlated tensors and no subsampling. Bottom Row: The plots are the same as for the top row, but zoomed out so that it is easier to see the behavior of alternating minimization after a large number of iterations.

We also ran both algorithms with noisy observations ($10\%$ noise entrywise). The error is measured with respect to the true tensor. We ran KRONECKER COMPLETION for $100$ iterations compared to running STANDARD ALTERNATING MINIMIZATION for $400$ iterations, to correct for the fact that each iteration of the former takes longer. Note (see Figure 2b) that the error achieved by both estimators is smaller than the noise that was added. Furthermore, the error decays with the square root of the number of samples, which is essentially optimal (see [6]). Although both algorithms converge to roughly the same error, similar to the non-noisy case, the error of our algorithm converges at a significantly faster rate in this setting as well.

Finally, we ran various tensor completion algorithms on correlated tensors for varying values of $n$ and numbers of observations. UNFOLDING involves unfolding the tensor and running alternating minimization for matrix completion on the resulting $n \times n^2$ matrix. We ran $100$ iterations for KRONECKER COMPLETION and UNFOLDING and $400$ iterations for STANDARD ALTERNATING MINIMIZATION. We defined success as achieving $< 1\%$ normalized MSE. Note how the sample complexity of KRONECKER COMPLETION appears to grow as $n^{3/2}$ whereas even for fairly large error tolerance ($1\%$), STANDARD ALTERNATING MINIMIZATION seems to have more difficulty scaling to larger tensors.

It is important to keep in mind that in many settings of practical interest we expect the factors to be correlated, such as in factor analysis [21]. Thus our experiments show that existing algorithms only work well in seriously restricted settings, and even then they exhibit various sorts of pathological behavior such as getting stuck without using subsampling, converging slower when there are more observations, etc. It appears that this sort of behavior only arises when completing tensors and not matrices. In contrast, our algorithm works well across a range of tensors (sometimes dramatically better) and fixes these issues.

**Additional plots**

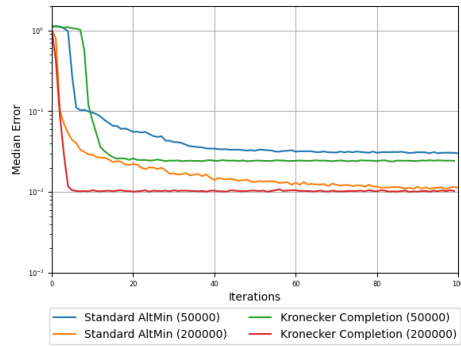
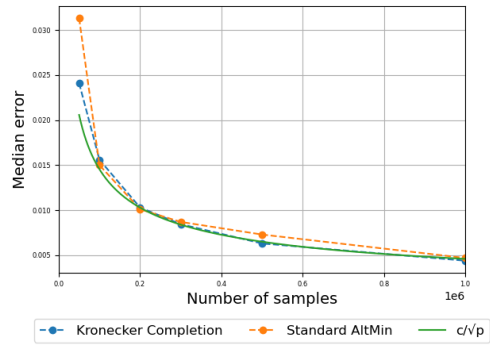

(a) Error over time with noisy observations, $n = 200, r = 4$

(b) Error vs samples with noisy observations, $n = 200, r = 4$.

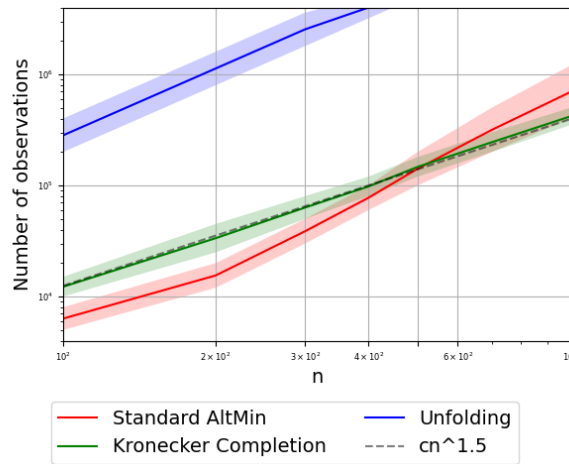

(c) Number of samples required for varying $n, r = 4$. Shaded areas represent $[20\%, 80\%]$ success rates.

Figure 2

# 5 Discussion

In this paper, we propose an algorithm for tensor completion that involves a modification to standard alternating minimization. We prove that our algorithm has strong theoretical guarantees. It runs in nearly linear time and provably achieves exact completion even for tensors whose components are highly correlated. Furthermore, we show empirically that the standard alternating minimization algorithm does get stuck in local minima when completing tensors with correlated components whereas our algorithm does not.

Moving forward, our work raises a few natural follow-up questions. From a theoretical perspective, our algorithm is limited to the under-complete case $r \leq n$ and our theoretical bounds scale somewhat poorly as a function of the rank. It would be fascinating to see if we can improve the dependence on $r$ or if our ideas can be extended to the overcomplete regime. From a practical perspective, there are several other existing algorithms beyond standard alternating minimization that would be interesting to compare with in empirical studies. These comparisons likely require a modified setup involving different types of tensors e.g. non-cubical or higher noise ratio. We leave these directions for future investigation

## Broader Impact

Our work gives new algorithms for tensor completion. These could certainly have positive societal benefit in terms of furnishing better methods to use in applications, such as those we discussed in the introduction. While a main application is to recommendation systems, where biases in the data can be amplified through algorithms, we believe that an important intermediate goal towards mitigating unfairness is to have a suite of algorithms where we can rigorously analyze them and understand their behavior. Our work takes a positive step in this direction.

## Acknowledgments and Disclosure of Funding

This work was supported in part by a Microsoft Trustworthy AI Grant, NSF CAREER Award CCF-1453261, NSF Large CCF1565235, a David and Lucile Packard Fellowship and an ONR Young Investigator Award

## Footnotes

\*This work was supported in part by a Microsoft Trustworthy AI Grant, NSF CAREER Award CCF-1453261, NSF Large CCF1565235, a David and Lucile Packard Fellowship, an Alfred P. Sloan Fellowship and an ONR Young Investigator Award.

[2]The Khatri-Rao product, which is less famililar than the Kronecker product, takes two matrices $A$ and $B$ with the same number of columns and forms a new matrix $C$ where the $i$th column of $C$ is the tensor product of the $i$th column of $A$ and the $i$th column of $B$. This operation has many applications in tensor analysis, particularly to prove (robust) identifiability results [1, 5].

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
