[Supplementary Material]

# Supplementary Material for "Tensor Completion Made Practical"

## A   Omitted algorithms from Section 3

Our initialization algorithm is based on [22]. For ease of notation, we make the following definition:

**Definition A.1.** *Let $\Pi$ be the projection map that projects an $n \times n$ matrix onto its diagonal entries and let $\Pi_\perp$ denote the projection onto the orthogonal complement.*

---

**Algorithm 3** INITIALIZATION

---

**Input:** Let $\widehat{T}$ be an input tensor where each entry is observed with probability $p_1$
Let $U = U_x(\widehat{T})$ be the unfolded tensor with all unobserved entries equal to 0
Define
$$\widehat{B} = \frac{1}{p_1}\Pi(UU^T) + \frac{1}{p_1^2}\Pi_\perp(UU^T)$$

Let $X$ be the matrix whose columns are given by the top $r$ singular vectors of $\widehat{B}$

Zero out all rows of $X$ that have norm at least $\tau\sqrt{\frac{r}{n}}$ where $\tau = \left(\frac{2\mu r}{c^2} \cdot \frac{\sigma_1^2}{\sigma_r^2}\right)^5$ and let $X_0$ be the resulting matrix
Let $V_x^0$ be the subspace spanned by the columns of $X_0$
Compute $V_y^0, V_z^0$ similarly using the corresponding unfoldings of $\widehat{T}$
**Output:** $V_x^0, V_y^0, V_z^0$

---

Now we present our algorithm for post-processing.

---

**Algorithm 4** POST-PROCESSING VIA CONVEX OPTIMIZATION

---

**Input:** Let $\widehat{T}$ be a sample where each entry is observed with probability $p_3$
**Input:** Let $\widehat{V_x}, \widehat{V_y}, \widehat{V_z}$ be subspace estimates that we are given
Split $\widehat{T}$ into two independent samples $\widehat{T_-}, \widehat{T_\sim}$ where each entry is observed with probability $p_3/2$
Let $S$ be the set of observed entries in $\widehat{T_-}$
Define:
$$T' = \arg\min_{T' \in \widehat{V_x} \otimes \widehat{V_y} \otimes \widehat{V_z}} \left\| (T' - \widehat{T_-})|_S \right\|_2^2$$

Run Jennrich's algorithm (see Section F.2.1) to decompose $T'$ into $r$ rank-1 components
$$T' = T_1 + \cdots + T_r$$

For each $1 \le i \le r$, write $T_i = \widehat{\sigma_i}\widehat{x_i} \otimes \widehat{y_i} \otimes \widehat{z_i}$ where $\widehat{x_i}, \widehat{y_i}, \widehat{z_i}$ are unit vectors and $\widehat{\sigma_i} \ge 0$.
Let $S_\sim$ be the set of observed entries in $\widehat{T_\sim}$
Solve the following constrained optimization problem where $a_1, b_1, c_1, \ldots, a_r, b_r, c_r \in \mathbb{R}^n$:

$$\min_{a_i, b_i, c_i} \left\| \left( T - \sum_{i=1}^r (\widehat{\sigma_i}(\widehat{x_i} + a_i)) \otimes (\widehat{y_i} + b_i) \otimes (\widehat{z_i} + c_i) \right)\Big|_{S_\sim} \right\|_2^2$$

over the polytope $Q(\widehat{x_1}, \ldots, \widehat{x_r}, \widehat{y_1}, \ldots, \widehat{y_r}, \widehat{z_1}, \ldots, \widehat{z_r})$ (defined below).
**Output:**
$$T_{\mathsf{est}} = \sum_{i=1}^r (\widehat{\sigma_i}(\widehat{x_i} + a_i)) \otimes (\widehat{y_i} + b_i) \otimes (\widehat{z_i} + c_i)$$

---

**Definition A.2** (Definition of $Q$). *For each $1 \le i \le r$, let $y_i'$ be the unit vector in $\mathsf{span}(\widehat{y_1}, \ldots, \widehat{y_r})$ that is orthogonal to $\widehat{y_1}, \ldots, \widehat{y_{i-1}}, \widehat{y_{i+1}}, \ldots, \widehat{y_r}$ and define $z_i'$ similarly. Let*
$$Q(\widehat{x_1}, \ldots, \widehat{x_r}, \widehat{y_1}, \ldots, \widehat{y_r}, \widehat{z_1}, \ldots, \widehat{z_r})$$

*be the polytope consisting of all $\{a_1, b_1, c_1, \ldots, a_r, b_r, c_r\}$ such that:*

- $0 \le ||a_i||_\infty, ||b_i||_\infty, ||c_i||_\infty \le \left(\frac{c\sigma_r}{10n\sigma_1}\right)^{10}$ *for all* $1 \le i \le r$.

- $b_i \cdot y_i' = 0$ *and* $c_i \cdot z_i' = 0$ *for all* $1 \le i \le r$.

**Remark.** *Note we will prove that the constrained optimization problem is strongly convex (and thus can be solved efficiently) in Section G.*

### A.1  "Exact" Completion and Bit Complexity

Technically, exact completion only makes sense when the entries of the hidden tensor have bounded bit complexity. Our algorithm achieves exact completion in the sense that, if we assume that all of the entries of the hidden tensor have bit complexity $B$, then the number of observations that our algorithm requires *does not* depend on $B$ while the runtime of our algorithm depends polynomially on $B$. This is the strongest possible guarantee one could hope for in the Word RAM model. Note that the last step of our algorithm involves solving a convex program. If we assume that the entries of the original tensor have bounded bit complexity then it suffices to solve the convex program to sufficiently high precision [12] and then round the solution.

## B  Outline of Proof of Theorem 3.2

Here we give an outline of the proof of Theorem 3.2. The first step involves proving that with high probability, the INITIALIZATION algorithm outputs subspaces that are incoherent and have constant principal angle with the true subspaces spanned by the unknown factors. The proof of the following theorem is in Section D.

**Theorem B.1.** *With probability* $1 - \frac{1}{n^{10}}$, *when the* INITIALIZATION *algorithm is run with*

$$p_1 = \left(\frac{2\mu r \log n}{c^2} \cdot \frac{\sigma_1^2}{\sigma_r^2}\right)^{10} \frac{1}{n^{3/2}},$$

*the output subspaces* $V_x^0, V_y^0, V_z^0$ *satisfy*

- $\max\left(\alpha(V_x, V_x^0), \alpha(V_y, V_y^0), \alpha(V_z, V_z^0)\right) \le 0.1$

- *The subspaces* $V_x^0, V_y^0, V_z^0$ *are* $\mu'$-*incoherent where* $\mu' = \left(\frac{2\mu r}{c^2} \cdot \frac{\sigma_1^2}{\sigma_r^2}\right)^{10}$

Next, we prove that each iteration of alternating minimization decreases the principal angles between our subspace estimates and the true subspaces. This is our main contribution.

**Theorem B.2.** *Consider the* KRONECKER ALTERNATING MINIMIZATION *algorithm. Fix a timestep $t$ and assume that the subspaces* $V_x^t, V_y^t, V_z^t$ *corresponding to the current estimate are* $\mu'$-*incoherent where* $\mu' = \left(\frac{2\mu r}{c^2} \cdot \frac{\sigma_1^2}{\sigma_r^2}\right)^{10}$. *Also assume*

$$\max\left(\alpha(V_x, V_x^t), \alpha(V_y, V_y^t), \alpha(V_z, V_z^t)\right) \le 0.1.$$

*If* $p' \ge \frac{\log^2((n\sigma_1)/(c\sigma_r))}{n^2}\left(\frac{10r\mu'}{c} \cdot \frac{\sigma_1}{\sigma_r}\right)^{10}$ *then after the next step, with probability* $1 - \frac{1}{10^4 \log((n\sigma_1)/(c\sigma_r))}$

- $\max\left(\alpha(V_x, V_x^{t+1}), \alpha(V_y, V_y^{t+1}), \alpha(V_z, V_z^{t+1})\right) \le 0.2 \max\left(\alpha(V_x, V_x^t), \alpha(V_y, V_y^t), \alpha(V_z, V_z^t)\right)$

- *The subspaces* $V_x^{t+1}, V_y^{t+1}, V_z^{t+1}$ *are* $\mu'$-*incoherent*

This theorem is proved in Section E

In light of the previous two theorems, by running a logarithmic number of iterations of alternating minimization, we can estimate the subspaces $V_x, V_y, V_z$ to within any inverse polynomial accuracy. This implies that we can estimate the entries of the true tensor $T$ to within any inverse polynomial

accuracy. A robust analysis of Jennrich's algorithm implies that we can then estimate the rank one components of the true tensor to within any inverse polynomial accuracy. Finally, since our estimates for the parameters $\widehat{\sigma}_i, \widehat{x}_i, \widehat{y}_i, \widehat{z}_i$ are close to the true parameters, we can prove that the optimization problem formulated in the POST-PROCESSING VIA CONVEX OPTIMIZATION algorithm is smooth and strongly convex. See Section F and Section G for details.

## C Preliminaries

### C.1 Basic Facts

We use the following notation:

- Let $U_x(T)$ denote the unfolding of $T$ into an $n \times n^2$ matrix where the dimension of length $n$ corresponds to the $x_i$. Define $U_y(T), U_z(T)$ similarly.
- Let $V_x$ be the subspace spanned by $x_1, \ldots, x_r$ and define $V_y, V_z$ similarly.
- Let $M_x$ be the matrix whose columns are $x_1, \ldots, x_r$ and define $M_y, M_z$ similarly.

The following claim, which states that the unfolded tensor $U_x(T)$ is not a degenerate matrix, will be used repeatedly later on.

**Claim C.1.** *The $r^{th}$ largest singular value of $U_x(T)$ is at least $c^3 \sigma_r$*

*Proof.* Let $D$ be the $r \times r$ diagonal matrix whose entries are $\sigma_1, \ldots, \sigma_r$. Let $N$ be the $r \times n^2$ matrix whose rows are $y_1 \otimes z_1, \ldots, y_r \otimes z_r$ respectively. Then

$$U_x(T) = M_x D N$$

For any unit vector $v \in V_x$, $\|v M_x\|_2 \geq c$. Also $N$ consists of a subset of the rows of $M_y \otimes M_z$ so the smallest singular value of $N$ is at least $c^2$. Thus

$$\|v U_x(T)\|_2 \geq c^3 \sigma_r.$$

Since $V_x$ has dimension $r$, this implies that the $r^{th}$ largest singular value of $U_x(T)$ is at least $c^3 \sigma_r$. □

A key component of our analysis will be tracking principal angles between subspaces. Intuitively, the principal angle between two $r$-dimensional subspaces $U, V \subset \mathbb{R}^n$ is the largest angle between some vector in $U$ and the subspace $V$.

**Definition C.2.** *For two subspaces $U, V \subset \mathbb{R}^n$ of dimension $r$, we let $\alpha(U, V)$ be the sine of the principal angle between $U$ and $V$. More precisely, if $U$ is a $n \times r$ matrix whose columns form an orthonormal basis for $U$ and $V_\perp$ is a $n \times (n - r)$ matrix whose columns form an orthonormal basis for the orthogonal complement of $V$, then*

$$\alpha(U, V) = \left\| V_\perp^T U \right\|_{op}$$

**Observation C.3** (Restatement of Observation 3.1). *Given subspaces $U_1, V_1 \subset \mathbb{R}^{n_1}$ of dimension $d_1$ and $U_2, V_2 \subset \mathbb{R}^{n_2}$ of dimension $d_2$, we have*

$$\alpha(U_1 \otimes U_2, V_1 \otimes V_2) \leq \alpha(U_1, V_1) + \alpha(U_2, V_2)$$

*Proof.* We slightly abuse notation and use $U_1, V_1, U_2, V_2$ to denote matrices whose columns form an orthonormal basis of the respective subspaces. Note that the cosine of the principal angle between $U_1$ and $V_1$ is equal to the smallest singular value of $U_1^T V_1$ and similar for $U_2$ and $V_2$. Next note

$$(U_1 \otimes U_2)^T (V_1 \otimes V_2) = (U_1^T V_1) \otimes (U_2^T V_2).$$

Thus

$$1 - \alpha(U_1 \otimes U_2, V_1 \otimes V_2)^2 = (1 - \alpha(U_1, V_1)^2)(1 - \alpha(U_2, V_2)^2)$$

and we conclude

$$\alpha(U_1 \otimes U_2, V_1 \otimes V_2) \leq \alpha(U_1, V_1) + \alpha(U_2, V_2)$$

□

In the analysis of our algorithm, we will also need to understand the incoherence of the tensor product of vector spaces. The following claim gives us a simple relation for this.

**Claim C.4.** *Suppose we have subspaces $V_1 \subset \mathbb{R}^{n_1}$ and $V_2 \subset \mathbb{R}^{n_2}$ with dimension $r_1, r_2$ that are $\mu_1$ and $\mu_2$ incoherent respectively. Then $V_1 \otimes V_2$ is $\mu_1\mu_2$-incoherent.*

*Proof.* Let $M_1, M_2$ be matrices whose columns are orthonormal bases for $V_1, V_2$ respectively. Then the columns of $M_1 \otimes M_2$ form an orthonormal basis for $V_1 \otimes V_2$. All rows of $M_1$ have norm at most $\sqrt{\frac{\mu_1 r_1}{n_1}}$ and all rows of $M_2$ have norm at most $\sqrt{\frac{\mu_2 r_2}{n_2}}$ so thus all rows of $M_1 \otimes M_2$ have norm at most $\sqrt{\frac{\mu_1 \mu_2 r_1 r_2}{n_1 n_2}}$ and we are done. $\qquad\square$

## C.2   Matrix and Incoherence Bounds

Here, we prove a few general results that we will use later to bound the principal angles and incoherence of the subspace estimates at each step of our algorithm.

**Claim C.5.** *Let $X$ be an $n \times m$ matrix such that:*

- *The rows of $X$ have norm at most $c\sqrt{\frac{r}{n}}$*

- *The $r^{th}$ singular value of $X$ is at least $\rho$*

*Then the subspace spanned by the top $r$ left singular vectors of $X$ is $\frac{c^2 r}{\rho^2}$-incoherent.*

*Proof.* Let the top $r$ left singular vectors of $X$ be $v_1, \ldots, v_r$. There are vectors $u_1, \ldots, u_r$ such that $v_i^T = Xu_i$ for all $i$. Furthermore, we can ensure

$$||u_i||_2 \le \frac{1}{\rho}$$

for all $i$. Now for a standard basis vector say $e_j$, its projection onto the subspace spanned by $v_1, \ldots, v_r$ has norm

$$\sqrt{(e_j^T Xu_1)^2 + \cdots + (e_j^T Xu_r)^2} \le \sqrt{r} \cdot c\sqrt{\frac{r}{n}}\frac{1}{\rho}.$$

Thus, the column space of $X$ is $\frac{c^2 r}{\rho^2}$-incoherent. $\qquad\square$

**Claim C.6.** *Let $A, B$ be $n \times m$ matrices with rank $r$. Let $\delta = ||A - B||_{op}$. Assume that the $r^{th}$ singular value of $B$ is at least $\rho$. Then the sine of the principal angle between the subspaces spanned by the columns of $A$ and $B$ is at most $\frac{\delta}{\rho}$.*

*Proof.* Let $V_A$ be the column space of $A$ and $V_B$ be the column space of $B$. Let $V_\perp$ be a $(n-r) \times n$ matrix whose rows form an orthonormal basis of the orthogonal complement of $V_A$. Note that

$$||V_\perp B|| \ge \rho\alpha(V_A, V_B)$$

Now $||V_\perp B|| \le ||V_\perp(B - A)|| \le \delta$. Thus $\alpha(V_A, V_B) \le \frac{\delta}{\rho}$ which completes the proof. $\qquad\square$

## C.3   Sampling Model

In our sampling model, we observe each entry of the tensor $T$ independently with probability $p$. In our algorithm we will require splitting the observations into several independent samples. To do this, we rely on the following claim:

**Claim C.7.** *Say we observe a sample $\widehat{T}$ where every entry of $T$ is revealed independently with probability $p$. Say $p_1 + p_2 \le p$. We can construct two independent samples $\widehat{T_1}, \widehat{T_2}$ where in the first sample, every entry is observed independently with probability $p_1$ and in the second, every entry is observed independently with probability $p_2$.*

*Proof.* For each entry in $\widehat{T}$ that we observe, reveal it in only $\widehat{T_1}$ with probability $\frac{p_1 - p_1 p_2}{p}$, reveal it in only $\widehat{T_2}$ with probability $\frac{p_2 - p_1 p_2}{p}$ and reveal it in both with probability $\frac{p_1 p_2}{p}$. Otherwise, don't reveal the entry in either $\widehat{T_1}$ or $\widehat{T_2}$. It can be immediately verified that $\widehat{T_1}$ and $\widehat{T_2}$ constructed in this way have the desired properties. $\square$

### C.4 Concentration Inequalities

**Claim C.8.** *Say we have real numbers $-\gamma \leq x_1, \ldots, x_n \leq \gamma$. Consider the sum*

$$X = \epsilon_1 x_1 + \cdots + \epsilon_n x_n$$

*where the $\epsilon_i$ are independent random variables that are equal to $1 - p$ with probability $p$ and equal to $-p$ with probability $1 - p$. Assume $p \geq \frac{1}{n}$. Then for any $t \geq 1$,*

$$\Pr\left[|X| \geq \sqrt{pn}\gamma t\right] \leq 2e^{-t/4}$$

*Proof.* Let $\beta$ be a constant with $0 \leq \beta \leq \gamma^{-1}$. Using the fact that $e^x \leq 1 + x + x^2$ between $-1$ and $1$, we have

$$\mathbb{E}[e^{\beta x_i}] = pe^{(1-p)\beta x_i} + (1-p)e^{-p\beta x_i} \leq 1 + p(1-p)\beta^2 x_i^2 \leq e^{p\beta^2 x_i^2}$$

Thus

$$\mathbb{E}[e^{\beta X}] \leq e^{p\beta^2(x_1^2 + \cdots + x_n^2)}$$

Similarly

$$\mathbb{E}[e^{-\beta X}] \leq e^{p\beta^2(x_1^2 + \cdots + x_n^2)}$$

Now set $\beta = \frac{1}{2\sqrt{pn}\gamma}$. Note that this is a valid assignment because we assumed $p \geq \frac{1}{n}$. Thus

$$\Pr\left[|X| \geq \sqrt{pn}\gamma t\right] \leq 2e^{p\beta^2 n\gamma^2 - \beta\sqrt{pn}\gamma t} \leq 2e^{-t/4}$$

$\square$

**Claim C.9.** *[Matrix Chernoff (see [28])] Consider a finite sequence $\{X_k\}$ of independent self-adjoint $n \times n$ matrices. Assume that for all $k$, $X_k$ is positive semidefinite and its largest eigenvalue is at most $R$. Let $\mu_{\min} = \lambda_{\min}\left(\sum_k \mathbb{E}[X_k]\right)$ be the smallest eigenvalue of the expected sum. Then*

$$\Pr\left[\lambda_{\min}\left(\sum_k X_k\right) \leq (1 - \delta)\mu_{\min}\right] \leq n e^{-\frac{\delta^2 \mu_{\min}}{2R}}$$

**Claim C.10.** *[Matrix Chernoff (see [28])] Consider a finite sequence $\{X_k\}$ of independent self-adjoint $n \times n$ matrices. Assume that for all $k$, $X_k$ is positive semidefinite and its largest eigenvalue is at most $R$. Let $\mu_{\max} = \lambda_{\max}\left(\sum_k \mathbb{E}[X_k]\right)$ be the largest eigenvalue of the expected sum. Then*

$$\Pr\left[\lambda_{\max}\left(\sum_k X_k\right) \leq (1 + \delta)\mu_{\max}\right] \leq n e^{-\frac{\delta^2 \mu_{\max}}{2R}}$$

## D  Initialization

The main purpose of this section is to prove that the INITIALIZATION algorithm obtains good initial estimates for the subspaces. In particular we will prove Theorem B.1. First note that

- $||U_x(T)||_{\mathsf{op}} \geq ||U_x(T)(y_1 \otimes z_1)||_2 = ||\sum_{i=1}^r \sigma_i((y_i \cdot y_1)(z_i \cdot z_1))x_i||_2 \geq \sigma_1 c$
- The largest norm of a row of $U_x(T)$ is at most $r\sigma_1 \sqrt{\frac{\mu r}{n}}$
- The largest norm of a column of $U_x(T)$ is at most $\sigma_1 \frac{\mu r^2}{n}$
- The largest entry of $U_x(T)$ is at most $r\sigma_1 \left(\frac{\mu r}{n}\right)^{3/2}$

Thus $U_x(T)$ is $(\lambda, 1, \rho)$-incoherent for $\lambda = \frac{\mu r^3}{c^2}, \rho = \frac{\mu^2 r^4}{c^2}$ according to Assumption 3 in [22]. Below we use $U_x$ to denote $U_x(T)$.

The key ingredient in the proof of Theorem B.1 is the following result from [22], which says that the matrix $\widehat{B}$ is a good approximation for $U_x U_x^T$.

**Lemma D.1** (Restatement of Corollary 3 in [22]). *When the* INITIALIZATION *algorithm is run with*

$$p_1 = \left( \frac{2\mu r \log n}{c^2} \cdot \frac{\sigma_1^2}{\sigma_r^2} \right)^{10} \frac{1}{n^{3/2}},$$

*we have*

$$||\widehat{B} - U_x U_x^T||_{op} \leq \frac{10^3 \lambda \rho (\log n)^4}{p_1^2 n^3} ||U_x||_{op}^2$$

*with probability at least* $1 - \left( \frac{1}{n} \right)^{25}$.

Let $D$ be the $r \times r$ diagonal matrix whose entries are the top $r$ signed eigenvalues of $\widehat{B}$. We first note that $\widehat{B}$ can be approximated well by its top $r$ principal components.

**Claim D.2.**

$$||XDX^T - \widehat{B}||_{op} \leq ||\widehat{B} - U_x U_x^T||_{op}$$

*Proof.* Let $\sigma$ be the $r + 1^{\text{st}}$ largest singular value of $\widehat{B}$. Note $||XDX^T - \widehat{B}||_{op} = \sigma$. Let $Y$ be the $r + 1$-dimensional space spanned by the top $r + 1$ singular vectors of $\widehat{B}$. Note that there is some unit vector $y \in Y$ such that $U_x^T y = 0$ (since $U_x^T$ has rank $r$) so

$$||\widehat{B} - U_x U_x^T||_{op} \geq ||(\widehat{B} - U_x U_x^T)y||_2 = ||\widehat{B}y||_2 \geq \sigma$$

$\square$

Let $\Pi$ be the $n \times n$ diagonal matrix with $0$ on diagonal entries corresponding to rows of $X$ with norm at least $\tau \sqrt{\frac{r}{n}}$ and $1$ on other diagonal entries. Note $X_0 = \Pi X$.

Lemma D.1 and Claim D.2 imply that $XDX^T$ is a good approximation for $U_x U_x^T$. To analyze the INITIALIZATION algorithm, we will need to rewrite these bounds using $X_0$ in place of $X$.

**Claim D.3.**

$$||X_0 D X_0^T - U_x U_x^T||_{op} \leq 0.1 \left( \frac{c\sigma_r}{\sigma_1} \right)^{10} \sigma_1^2$$

*with probability at least* $1 - \frac{1}{n^{25}}$

*Proof.* First note that the squared Frobenius norm of $X$ is $r$ so there are at most $\frac{n}{\tau^2}$ entries of $\Pi$ that are $0$. Next note

$$X_0 D X_0^T - U_x U_x^T = ((\Pi X)D(\Pi X)^T - (\Pi U_x)(\Pi U_x)^T) + (\Pi U_x)(\Pi U_x)^T - U_x U_x^T$$

Note that $(\Pi U_x)(\Pi U_x)^T - U_x U_x^T$ is $0$ in all but at most $\frac{2n^2}{\tau^2}$ entries. Furthermore, all entries of $U_x U_x^T$ are at most $\frac{\mu^3 \sigma_1^2 r^5}{n}$. Thus

$$||(\Pi U_x)(\Pi U_x)^T - U_x U_x^T||_2 \leq \frac{\mu^3 \sigma_1^2 r^5}{n} \cdot \frac{2n}{\tau} = \frac{2\mu^3 \sigma_1^2 r^5}{\tau}$$

Also

$$||(\Pi X)D(\Pi X)^T - (\Pi U_x)(\Pi U_x)^T)||_{op} \leq ||XDX^T - U_x U_x^T||_{op} \leq 2||\widehat{B} - U_x U_x^T||_{op} \leq \frac{20^3 \lambda \rho (\log n)^4}{p_0^2 n^3} ||U_x||_{op}^2$$

where we used Claim D.2 and Lemma D.1. Combining the previous two equations and noting that $||U_x||_{op} \leq (\sigma_1 r)$

$$||X_0 D X_0^T - U_x U_x^T||_{op} \leq \frac{2\mu^3 \sigma_1^2 r^5}{\tau} + \frac{20^3 \lambda \rho (\log n)^4}{p_0^2 n^3} (\sigma_1 r)^2 \leq 0.1 \left( \frac{c\sigma_r}{\sigma_1} \right)^{10} \sigma_1^2$$

$\square$

To prove that $V_x^0$, the space spanned by the columns of $X_0$ is incoherent, we will need a bound on the smallest singular value of $X_0$. In other words, we need to ensure that zeroing out the rows of $X$ whose norm is too large doesn't degenerate the column space.

**Claim D.4.** *With probability* $1 - \frac{1}{n^{25}}$, *the $r^{th}$ singular value of $X_0$ is at least* $\frac{c^6}{10r^3} \left( \frac{\sigma_r}{\sigma_1} \right)^2$

*Proof.* By Claim C.1, the $r^{\text{th}}$ singular value of $U_x U_x^T$ is at least $c^6 \sigma_r^2$. Thus by Claim D.3, the $r^{\text{th}}$ singular value of $X_0 D X_0^T$ is at least $0.9 c^6 \sigma_r^2$.

Also note that the operator norm of $U_x U_x^T$ is at most $(\sigma_1 r)^2$ so by Lemma D.1, all entries of $D$ are at most $2(\sigma_1 r)^2$. Also, the largest singular value of $X_0$ is clearly at most $r$. Thus the smallest singular value of $X_0$ is at least

$$\frac{0.9 c^6 \sigma_r^2}{2(\sigma_1 r)^2 r} \geq \frac{c^6}{10r^3} \left( \frac{\sigma_r}{\sigma_1} \right)^2$$

$\square$

Now we can use the results from Section C.2 to complete the proof of Theorem B.1.

**Corollary D.5.** *With probability at least* $1 - \left( \frac{1}{n} \right)^{25}$, *the subspace spanned by the columns of $X_0$ is* $\left( \frac{2\mu r}{c^2} \cdot \frac{\sigma_1^2}{\sigma_r^2} \right)^{10}$-*incoherent.*

*Proof.* Note all rows of $X_0$ have norm at most $\tau \sqrt{\frac{r}{n}}$. By Claim C.5 and Claim D.4, the column space of $X_0$ is incoherent with incoherence parameter

$$\frac{\tau^2 r}{\left( \frac{c^6}{10r^3} \left( \frac{\sigma_r}{\sigma_1} \right)^2 \right)^2} \leq \left( \frac{2\mu r}{c^2} \cdot \frac{\sigma_1^2}{\sigma_r^2} \right)^{10} .$$

$\square$

**Corollary D.6.** *With probability at least* $1 - \frac{1}{n^{25}}$ *we have* $\alpha(V_x, V_x^0) \leq 0.1$.

*Proof.* Note that $X_0 D X_0^T$ and $U_x U_x^T$ are both $n \times n$ matrices with rank $r$. By Claim C.1, the $r^{\text{th}}$ singular value of $U_x$ is at least $c^3 \sigma_r$ and thus the $r^{\text{th}}$ singular value of $U_x U_x^T$ is at least $c^6 \sigma_r^2$. Now using Claim D.3 and Claim C.6, we have with probability at least $1 - \frac{1}{n^{25}}$

$$\alpha(V_x, V_x^0) \leq \frac{0.1 \left( \frac{c\sigma_r}{\sigma_1} \right)^{10} \sigma_1^2}{c^6 \sigma_r^2} \leq 0.1 .$$

$\square$

*Proof of Theorem B.1.* Combining Corollary D.5 and Corollary D.6 we immediately get the desired.

$\square$

# E    Alternating Minimization

This section is devoted to the proof of Theorem B.2.

## E.1    Least Squares Optimization

We will need to analyze the least squares optimization in the alternating minimization step of our KRONECKER ALTERNATING MINIMIZATION algorithm. We use the following notation.

- Let the rows of $H_{t+1}$ be $r_1, \dots, r_n$.

- We will abbreviate $U_x(T)$ with $U_x$. Let the rows of $U_x(T)$ be $u_1, \ldots, u_n$.
- For each $1 \leq j \leq n$, let $P_j \subset [n^2]$ be the set of indices that are revealed in the $j^{\text{th}}$ row of $U_x(\widehat{T}_{t+1})$.
- Let $\Pi_j$ be the $n^2 \times n^2$ matrix with one in the diagonal entries corresponding to elements of $P_j$ and zero everywhere else.

Let $E = H_{t+1} - U_x B_t^T$ and let its rows be $s_1, \ldots, s_n$. Note $U_x B_t^T$ is the solution to the least squares optimization problem if we were able to observe all the entries i.e. if the optimization problem were not restricted to the set $S_{t+1}$. Thus, we can think of $E$ as the error that comes from only observing a subset of the entries. The end goal of this section is to bound the Frobenius norm of $E$.

First we will need a technical claim that follows from a matrix Chernoff bound.

**Claim E.1.** *If* $p' \geq \frac{\log^2 n}{n^2} \left( \frac{10r\mu'}{c} \cdot \frac{\sigma_1}{\sigma_r} \right)^{10}$ *then With at least* $1 - \left( \frac{c\sigma_r}{n\sigma_1} \right)^{20}$ *probability*

$$||(B_t \Pi_j B_t^T)^{-1}|| \leq \frac{2}{p'}$$

*for all* $1 \leq j \leq n$.

*Proof.* Note by Claim C.4, the columns of $B_t$ have norm at most $\frac{\mu' r}{n}$. Consider $\frac{1}{p'} B_t \Pi_j B_t^T$. This is a sum of independent rank 1 matrices with norm bounded by $\frac{\mu'^2 r^2}{p' n^2}$ and the expected value of the sum is $I$, the identity matrix. Thus by Claim C.9

$$\Pr\left[ \lambda_{\min} \left( \frac{1}{p'} B_t \Pi_j B_t^T \right) \leq \frac{1}{2} \right] \leq n e^{-\frac{p' n^2}{8\mu'^2 r^2}} \leq \left( \frac{c\sigma_r}{n\sigma_1} \right)^{25}$$

$\square$

The claim below follows directly from writing down the explicit formula for the solution to the least squares optimization problem.

**Claim E.2.** *We have for all* $1 \leq j \leq n$

$$r_j = u_j \Pi_j B_t^T (B_t \Pi_j B_t^T)^{-1}$$

*Proof.* Note $r_j$ must have the property that the vector $u_j - r_j B_t$ restricted to the entries indexed by $P_j$ is orthogonal to the space spanned by the rows of $B_t$ restricted to the entries indexed by $P_j$. This means

$$(u_j - r_j B_t)(B_t \Pi_j)^T = 0$$

The above rearranges as

$$r_j(B_t \Pi_j B_t^T) = u_j \Pi_j B_t^T$$

from which we immediately obtain the desired.

$\square$

After some direct computations, the previous claim implies:

**Claim E.3.** *We have for all* $1 \leq j \leq n$

$$s_j = u_j(I - B_t^T B_t)\Pi_j B_t^T (B_t \Pi_j B_t^T)^{-1}$$

*Proof.* Note

$$s_j = r_j - u_j B_t^T = u_j \Pi_j B_t^T (B_t \Pi_j B_t^T)^{-1} - u_j B_t^T = u_j(I - B_t^T B_t)\Pi_j B_t^T (B_t \Pi_j B_t^T)^{-1}$$

$\square$

Now we are ready to prove the main result of this section.

**Lemma E.4.** *With probability at least* $1 - \frac{1}{10^5 \log((n\sigma_1)/(c\sigma_r))}$

$$||E||_2 \leq \frac{10^3 \mu' r^3 \sigma_1 \sqrt{\log((n\sigma_1)/(c\sigma_r))}}{n\sqrt{p'}} \left( \alpha(V_y, V_y^t) + \alpha(V_z, V_z^t) \right)$$

*Proof.* Note $B_t B_t^T = I$ since its rows form an orthonormal basis so $u_j(I - B_t^T B_t)B_t^T = 0$. Thus

$$s_j = u_j(I - B_t^T B_t)(\Pi_j - p'I)B_t^T(B_t\Pi_j B_t^T)^{-1}$$

Let $q_j = u_j(I - B_t^T B_t)$. By Observation 3.1,

$$||q_j||_2 \leq ||u_j||_2 \alpha(V_y \otimes V_z, V_y^t \otimes V_z^t) \leq ||u_j||_2 \left(\alpha(V_y, V_y^t) + \alpha(V_z, V_z^t)\right) \tag{4}$$

Now we upper bound $||q_j(\Pi_j - p'I)B_t^T||$. Let the columns of $B_t^T$ be $c_1, \ldots, c_{r^2}$ respectively. Note that since the entries of $\Pi_j - p'I$ have expectation 0 and variance $p'(1-p')$

$$\mathbb{E}[||q_j(\Pi_j - p'I)c_i||_2^2] \leq ||q_j||_2^2 \cdot p'(1-p') \cdot ||c_i||_\infty^2 \leq \frac{p'||q_j||_2^2 \mu'^2 r^2}{n^2}$$

Thus

$$\mathbb{E}[||q_j(\Pi_j - p'I)B_t^T||_2^2] \leq \frac{p'||q_j||_2^2 \mu'^2 r^4}{n^2} \tag{5}$$

Now by Claim E.1, with probability $1 - \left(\frac{c\sigma_r}{n\sigma_1}\right)^{20}$ we have

$$||(B_t\Pi_j B_t^T)^{-1}|| \leq \frac{2}{p'}$$

for all $1 \leq j \leq n$. Denote this event by $\Gamma$. Let

$$\mathbb{E}_\Gamma\left[||E||_2^2\right] = \Pr\left[\Gamma\right]\mathbb{E}\left[||E||_2^2 \mid \Gamma\right]$$

In other words, $\mathbb{E}_\Gamma[||E||_2^2]$ is the sum of the contribution to $\mathbb{E}\left[||E||_2^2\right]$ from events where $\Gamma$ happens.

Using (4) and (5) we have

$$\mathbb{E}_\Gamma\left[||E||_2^2\right] = \sum_{j=1}^n \mathbb{E}_\Gamma\left[||s_j||_2^2\right] \leq \frac{4}{p'^2}\frac{p'\mu'^2 r^4}{n^2}\left(\sum_{j=1}^n ||q_j||_2^2\right)$$

$$\leq \frac{4\mu'^2 r^4}{p'n^2}\left(\alpha(V_y, V_y^t) + \alpha(V_z, V_z^t)\right)^2 \sum_{j=1}^n ||u_j||_2^2$$

$$\leq \frac{4\mu'^2 r^6 \sigma_1^2}{p'n^2}\left(\alpha(V_y, V_y^t) + \alpha(V_z, V_z^t)\right)^2$$

By Markov's inequality, the probability that $\Gamma$ occurs and

$$||E||_2 \geq \frac{10^3 \mu' r^3 \sigma_1 \sqrt{\log((n\sigma_1)/(c\sigma_r))}}{n\sqrt{p'}}\left(\alpha(V_y, V_y^t) + \alpha(V_z, V_z^t)\right)$$

is at most $\frac{1}{2 \cdot 10^5 \log((n\sigma_1)/(c\sigma_r))}$. Thus with probability at least

$$1 - \frac{1}{2 \cdot 10^5 \log((n\sigma_1)/(c\sigma_r))} - \left(\frac{c\sigma_r}{n\sigma_1}\right)^{20} \geq 1 - \frac{1}{10^5 \log((n\sigma_1)/(c\sigma_r))}$$

we have

$$||E||_2 \leq \frac{10^3 \mu' r^3 \sigma_1 \sqrt{\log((n\sigma_1)/(c\sigma_r))}}{n\sqrt{p'}}\left(\alpha(V_y, V_y^t) + \alpha(V_z, V_z^t)\right)$$

$\square$

We will need one additional lemma to show that the incoherence of the subspaces is preserved. This lemma is an upper bound on the norm of the rows of $E$. Note this upper bound is much weaker than the one in the previous lemma and does not capture the progress made by alternating minimization but rather is a fixed upper bound that holds for all iterations.

**Lemma E.5.** *If $p' \geq \frac{\log^2((n\sigma_1)/(c\sigma_r))}{n^2}\left(\frac{10r\mu'}{c} \cdot \frac{\sigma_1}{\sigma_r}\right)^{10}$ then with probability at least $1 - \left(\frac{c\sigma_r}{n\sigma_1}\right)^{15}$ all rows of $E$ have norm at most $\frac{0.1c^3\sigma_r}{\sqrt{n}}$*

*Proof.* We use the same notation as the proof of the previous claim. Fix indices $1 \leq j \leq n, 1 \leq i \leq r^2$. It will suffice to obtain a high probability bound for

$$|q_j(\Pi_j - p'I)c_i|$$

and then union bound over all indices. Recall

$$q_j = u_j - u_j B_t^T B_t$$

and by our incoherence assumptions, all entries of $u_j$ have magnitude at most $r\sigma_1 \left(\frac{\mu r}{n}\right)^{3/2}$. By Claim C.4, all entries of $u_j B_t^T B_t$ have magnitude at most

$$||u_j B_t^T||_2 \frac{\mu'r}{n} \leq ||u_j||_2 \frac{\mu'r}{n} \leq r\sigma_1 \sqrt{\frac{\mu r}{n}} \frac{\mu'r}{n}$$

Thus all entries of $q_j$ have magnitude at most $2r\sigma_1 \sqrt{\frac{\mu r}{n}} \frac{\mu'r}{n}$. Let $\tau$ be the vector obtained by taking the entrywise product of $q_j$ and $c_i$ and say its entries are $\tau_1, \ldots, \tau_{n^2}$. Note that by Claim C.4 the entries of $c_i$ are all at most $\frac{\mu'r}{n}$. Thus the entries of $\tau$ are all at most

$$\gamma = 2r\sigma_1 \sqrt{\frac{\mu r}{n}} \frac{\mu'^2 r^2}{n^2}$$

Next observe that $q_j(\Pi_j - p'I)c_i$ is obtained by taking a sum $\epsilon_1\tau_1 + \cdots + \epsilon_{n^2}\tau_{n^2}$ where $\epsilon_1, \ldots, \epsilon_{n^2}$ are sampled independently and are equal to $-p'$ with probability $1 - p'$ and equal to $1 - p'$ with probability $p'$. We now use Claim C.8. Note $p' \geq \frac{1}{n^2}$ clearly. Thus

$$\Pr\left[|q_j(\Pi_j - p'I)c_i| \geq 10^3 \log((n\sigma_1)/(c\sigma_r))n\gamma\sqrt{p'}\right] \leq \left(\frac{c\sigma_r}{n\sigma_1}\right)^{25}.$$

Note

$$n\gamma\sqrt{p'} = \frac{2r^{3.5}\mu'^2\mu^{0.5}\sqrt{p'}\sigma_1}{n^{3/2}}$$

Thus

$$\Pr\left[|q_j(\Pi_j - p'I)c_i| \leq \frac{2 \cdot 10^3 \log((n\sigma_1)/(c\sigma_r))r^{3.5}\mu'^2\mu^{0.5}\sqrt{p'}\sigma_1}{n^{3/2}}\right] \geq 1 - \left(\frac{c\sigma_r}{n\sigma_1}\right)^{25}.$$

Union bounding over $1 \leq i \leq r^2$ implies that with probability at least $1 - \left(\frac{c\sigma_r}{n\sigma_1}\right)^{23}$

$$||q_j(\Pi_j - p'I)B_t^T|| \leq \frac{2 \cdot 10^3 \log((n\sigma_1)/(c\sigma_r))r^{4.5}\mu'^2\mu^{0.5}\sqrt{p'}\sigma_1}{n^{3/2}}$$

Finally, since

$$s_j = u_j(I - B_t^T B_t)(\Pi_j - p'I)B_t^T(B_t\Pi_j B_t^T)^{-1} = q_j(\Pi_j - p'I)B_t^T(B_t\Pi_j B_t^T)^{-1}$$

combining with Claim E.1 and union bounding over all $1 \leq j \leq n$, we see that with at least $1 - \left(\frac{c\sigma_r}{n\sigma_1}\right)^{15}$ probability, all rows of $E$ have norm at most

$$\frac{4 \cdot 10^3 \log((n\sigma_1)/(c\sigma_r))r^{4.5}\mu'^2\mu^{0.5}\sigma_1}{n^{3/2}\sqrt{p'}} \leq \frac{0.1c^3\sigma_r}{\sqrt{n}}$$

which completes the proof. $\square$

## E.2 Progress Measure

Now we will use the bounds in Lemma E.4 and Lemma E.5 on the error term $E$ to bound the principal angle with respect to $V_x$ and the incoherence of the new subspace estimate $V_x^{t+1}$. This will then complete the proof of Theorem B.2. We first need a preliminary result controlling the $r^{\text{th}}$ singular value of $U_x B_t^T$. Note this is necessary because if the $r^{\text{th}}$ singular value of $U_x B_t^T$ were too small, then it could be erased by the error term $E$.

**Claim E.6.** *The $r^{th}$ largest singular value of $U_x B_t^T$ is at least $\left(1 - \alpha(V_y, V_y^t) - \alpha(V_z, V_z^t)\right) c^3 \sigma_r$*

*Proof.* By Claim C.1, the $r^{\text{th}}$ largest singular value of $U_x$ is at least $c^3 \sigma_r$. Therefore there exists an $r$-dimensional subspace of $\mathbb{R}^n$, say $V$ such that for any unit vector $v \in V$, $||vU_x|| \geq c^3 \sigma_r$. Now for any vector $u$ in $V_y \otimes V_z$,

$$||uB_t^T|| \geq \sqrt{1 - \alpha(V_y \otimes V_z, V_y^t \otimes V_z^t)^2} ||u||$$

Next $vU_x$ is contained in the row span of $U_x$ which is contained in $V_y \otimes V_z$. Thus for any unit vector $v \in V$

$$||vU_x B_t^T|| \geq c^3 \sigma_r \sqrt{1 - \alpha(V_y \otimes V_z, V_y^t \otimes V_z^t)^2} \geq \left(1 - \alpha(V_y, V_y^t) - \alpha(V_z, V_z^t)\right) c^3 \sigma_r$$

$\square$

Now we can upper bound the principal angle between $V_x$ and $V_x^{t+1}$ in terms of the principal angles for the previous iterates.

**Corollary E.7.** *If $\quad p' \quad \geq \quad \frac{\log^2((n\sigma_1)/(c\sigma_r))}{n^2} \left(\frac{10r\mu'}{c} \cdot \frac{\sigma_1}{\sigma_r}\right)^{10} \quad$ then $\quad \alpha(V_x^{t+1}, V_x) \quad \leq$ $0.1 \left(\alpha(V_y, V_y^t) + \alpha(V_z, V_z^t)\right)$ with at least $1 - \frac{1}{10^5 \log((n\sigma_1)/(c\sigma_r))}$ probability.*

*Proof.* Note $H = U_x B_t^T + E$.
By Lemma E.4, with probability at least $1 - \frac{1}{10^5 \log((n\sigma_1)/(c\sigma_r))}$

$$||E||_2 \leq \frac{10^3 \mu' r^3 \sigma_1 \sqrt{\log(n/(c\sigma_r))}}{n\sqrt{p'}} \left(\alpha(V_y, V_y^t) + \alpha(V_z, V_z^t)\right)$$

If this happens, the largest singular value of $E$ is at most

$$\sigma \leq ||E||_2 \leq \frac{10^3 \mu' r^3 \sigma_1 \sqrt{\log(n/(c\sigma_r))}}{n\sqrt{p'}} \left(\alpha(V_y, V_y^t) + \alpha(V_z, V_z^t)\right)$$

Let $\rho_1 \geq \cdots \geq \rho_r$ be the singular values of $U_x B_t^T$. By Claim E.6

$$\rho_r \geq \left(1 - \alpha(V_y, V_y^t) - \alpha(V_z, V_z^t)\right) c^3 \sigma_r$$

Let $H_r$ be the rank-$r$ approximation of $H$ given by the top $r$ singular components. $U_x B_t^T$ has rank $r$ and since $H_r$ is the best rank $r$ approximation of $H$ in Frobenius norm we have $H_r = H + E' = U_x B_t^T + E + E'$ where $||E'||_2 \leq ||E||_2$. Now note

$$\left\| H_r - U_x B_t^T \right\|_{\text{op}} \leq ||E + E'||_2 \leq 2 ||E||_2$$

Thus by Claim C.6 (applied to the matrices $H_r$, $U_x B_t^T$)

$$\alpha(V_x, V_x^{t+1}) \leq \frac{2||E||_2}{\rho_r} \leq \frac{4||E||_2}{\sigma_r c^3} \leq 0.1 \left(\alpha(V_y, V_y^t) + \alpha(V_z, V_z^t)\right)$$

$\square$

We also upper bound the incoherence of $V_x^{t+1}$, relying on Lemma E.5.

**Corollary E.8.** *If $p' \geq \frac{\log^2((n\sigma_1)/(c\sigma_r))}{n^2} \left(\frac{10r\mu'}{c} \cdot \frac{\sigma_1}{\sigma_r}\right)^{10}$ then the subspace $V_x^{t+1}$ is $\left(\frac{2\mu r}{c^2} \cdot \frac{\sigma_1^2}{\sigma_r^2}\right)^{10}$-incoherent with at least $1 - \left(\frac{c\sigma_r}{n\sigma_1}\right)^{10}$ probability.*

*Proof.* By Lemma E.5, with $1 - \left(\frac{c\sigma_r}{n\sigma_1}\right)^{10}$ probability, each row of $E$ has norm at most $\frac{0.1c^3\sigma_r}{\sqrt{n}}$. This implies $||E||_2 \leq 0.1c^3\sigma_r$.

Let the top $r$ singular values of $H$ be $\rho'_1, \dots, \rho'_r$ and let the top $r$ singular values of $U_x B_t^T$ be $\rho_1, \dots, \rho_r$. Note $\rho'_r \geq \rho_r - \|E\|_{\text{op}}$. Also by Claim E.6

$$\rho_r \geq \left(1 - \alpha(V_y, V_y^t) - \alpha(V_z, V_z^t)\right) c^3 \sigma_r$$

Thus

$$\rho'_r \geq \rho_r - \|E\|_{\text{op}} \geq \left(1 - \alpha(V_y, V_y^t) - \alpha(V_z, V_z^t)\right) c^3 \sigma_r - \|E\|_2 \geq \frac{c^3 \sigma_r}{2}$$

Next observe that each row of $U_x B_t^T$ has norm at most $r\sigma_1 \sqrt{\frac{\mu r}{n}}$ and since each row of $E$ has norm at most $\frac{0.1 c^3 \sigma_r}{\sqrt{n}}$, we deduce that each row of $H$ has norm at most $2r\sigma_1 \sqrt{\frac{\mu r}{n}}$. Now by Claim C.5, $V_x^{t+1}$ is incoherent with incoherence parameter

$$\frac{4r^3 \mu \sigma_1^2}{\left(\frac{c^3 \sigma_r}{2}\right)^2} = \frac{16 \mu r^3 \sigma_1^2}{c^6 \sigma_r^2}.$$

Clearly

$$\frac{16 \mu r^3 \sigma_1^2}{c^6 \sigma_r^2} \leq \left(\frac{2\mu r}{c^2} \cdot \frac{\sigma_1^2}{\sigma_r^2}\right)^{10}$$

so we are done. $\qquad\square$

We can now complete the proof of the main theorem of this section, Theorem B.2.

*Proof of Theorem B.2.* Combining Corollary E.7 and Corollary E.8, we immediately get the desired.
$\qquad\square$

Theorem B.2 immediately gives us the following corollary.

**Corollary E.9.** *Assume that* $V_x^0, V_y^0, V_z^0$ *satisfy*

- $\max\left(\alpha(V_x, V_x^0), \alpha(V_y, V_y^0), \alpha(V_z, V_z^0)\right) \leq 0.1$

- *The subspaces* $V_x^0, V_y^0, V_z^0$ *are* $\mu'$-*incoherent where* $\mu' = \left(\frac{2\mu r}{c^2} \cdot \frac{\sigma_1^2}{\sigma_r^2}\right)^{10}$

*Then with probability* $0.99$, *when* KRONECKER ALTERNATING MINIMIZATION *is run with parameters*

$$p_2 = \left(\frac{\mu r \log n}{c} \cdot \frac{\sigma_1}{\sigma_r}\right)^{300} \frac{1}{n^{3/2}}$$

$$p' = \frac{\log^2((n\sigma_1)/(c\sigma_r))}{n^2} \left(\frac{10 r \mu'}{c} \cdot \frac{\sigma_1}{\sigma_r}\right)^{10}$$

*we have*

$$\max\left(\alpha(V_x, V_x^k), \alpha(V_y, V_y^k), \alpha(V_z, V_z^k)\right) \leq \left(\frac{\sigma_r c}{10 \sigma_1 n}\right)^{10^2}.$$

## F  Projection and Decomposition

After computing estimates for $V_x, V_y, V_z$, say $\widehat{V_x}, \widehat{V_y}, \widehat{V_z}$, we estimate the original tensor by projecting onto the space spanned by $\widehat{V_x} \otimes \widehat{V_y} \otimes \widehat{V_z}$. Our first goal is to show that our error in estimating $T$ by projecting onto these estimated subspaces depends polynomially on the principal angles $\alpha(V_x, \widehat{V_x}), \alpha(V_y, \widehat{V_y}), \alpha(V_z, \widehat{V_z})$. This will then imply (due to Theorem B.2) that we can estimate the entries of the original tensor to any inverse polynomial accuracy.

## F.1 Projection Step

We will slightly abuse notation and use $\widehat{V_x}, \widehat{V_y}, \widehat{V_z}$ to denote $n \times r$ matrices whose columns form orthonormal bases of the respective subspaces. Let

$$\delta = \max\left(\alpha(V_x, \widehat{V_x}), \alpha(V_y, \widehat{V_y}), \alpha(V_z, \widehat{V_z})\right)$$

Let $M = V_x \otimes V_y \otimes V_z$ be an $r^3 \times n^3$ matrix. Let $S$ be a subset of the rows of $M$ where each row is chosen with probability $p' = \frac{\log^2 n}{n^2}\left(\frac{10r\mu}{c} \cdot \frac{\sigma_1}{\sigma_r}\right)^{10}$. Let $M_S$ be the matrix obtained by taking only the rows of $M$ in $S$.

The main lemma of this section is:

**Lemma F.1.** *Assume that $\delta \leq \left(\frac{c\sigma_r}{n\sigma_1}\right)^{10}$. When the* POST-PROCESSING VIA CONVEX OPTIMIZA-TION *algorithm is run with $p_3 = 2\frac{\log^2 n}{n^2}\left(\frac{10r\mu}{c} \cdot \frac{\sigma_1}{\sigma_r}\right)^{10}$, the tensor $T'$ satisfies*

$$||T - T'||_2 \leq 10\sigma_1 \delta r n$$

*with probability at least $1 - \frac{1}{n^{20}}$.*

We first prove a preliminary claim.

**Claim F.2.** *Assume that $\delta \leq \left(\frac{c\sigma_r}{n\sigma_1}\right)^{10}$. Then with probability at least $1 - \frac{1}{n^{20}}$, the smallest singular value of $M_S$ is at least $\frac{1}{n}$*

*Proof.* First we show that all of the entries of $\widehat{V_x}$ are at most $2\sqrt{\frac{\mu r}{n}}$ in magnitude. Assume for the sake of contradiction that some entry of $\widehat{V_x}$ is at least $2\sqrt{\frac{\mu r}{n}}$. Let $c$ be the column of $\widehat{V_x}$ that contains this entry. Let $\Pi(c)$ be the projection of $c$ onto the subspace $V_x$. Note that all entries of $\Pi(c)$ are at most $\sqrt{\frac{\mu r}{n}}$ since $V_x$ is $\mu$-incoherent. This means that

$$||c - \Pi(c)||_2^2 \geq \frac{\mu r}{n}$$

so

$$\alpha(\widehat{V_x}, V_x) \geq \sqrt{\frac{\mu r}{n}}$$

contradicting the assumption that $\alpha(V_x, \widehat{V_x}) \leq \left(\frac{c\sigma_r}{n\sigma_1}\right)^{10}$. Similarly, we get the same bound for the entries of $\widehat{V_y}, \widehat{V_z}$. This implies each row of $M$ has norm at most $\frac{8\mu^{3/2}r^3}{n^{3/2}}$. Now consider $\frac{1}{p'}M_S^T M_S$. This is a sum of independent rank-1 matrices with norm at most $\frac{1}{p'} \cdot \frac{64\mu^3 r^6}{n^3}$ and the expected value of the sum is $I$, the identity matrix. Thus by Claim C.9

$$\Pr\left[\lambda_{\min}\left(\frac{1}{p'}M_S^T M_S\right) \leq \frac{1}{2}\right] \leq \frac{1}{n^{20}}$$

In particular, with probability at least, $1 - \frac{1}{n^{20}}$, the smallest singular value of $M_S$ is at least

$$\sqrt{\frac{p'}{2}} \geq \frac{1}{n}$$

$\square$

*Proof of Lemma F.1.* For $1 \leq i \leq r$, let $\overline{x_i}$ be the projection of $x_i$ onto $\widehat{V_x}$. Define $\overline{y_i}, \overline{z_i}$ similarly. Let

$$\overline{T} = \sum_{i=1}^{r} \sigma_i \overline{x_i} \otimes \overline{y_i} \otimes \overline{z_i}$$

Note

$$||\overline{x_i} \otimes \overline{y_i} \otimes \overline{z_i} - x_i \otimes y_i \otimes z_i||_2 \le ||\overline{x_i} - x_i||_2 + ||\overline{y_i} - y_i||_2 + ||\overline{z_i} - z_i||_2$$
$$\le \alpha(V_x, \widehat{V_x}) + \alpha(V_y, \widehat{V_y}) + \alpha(V_z, \widehat{V_z})$$
$$\le 3\delta$$

Thus

$$||\overline{T} - T||_2 \le 3\delta\sigma_1 r \tag{6}$$

Now consider the difference $\overline{T} - T'$. By the definition of $T'$ We must have

$$||(T' - T)|_S||_2 \le \left||(\overline{T} - T)|_S\right||_2 \le 3\delta\sigma_1 r$$

Thus $\left||(T' - \overline{T})|_S\right||_2 \le 6\delta\sigma_1 r$. Since $T', \overline{T}$ are both in the subspace $\widehat{V_x} \otimes \widehat{V_y} \otimes \widehat{V_z}$, when flattened $T', \overline{T}$ can be written in the form $Mv', M\overline{v}$ respectively for some $v', \overline{v} \in \mathbb{R}^{r^3}$. Now we know

$$||M_S(v' - \overline{v})||_2 \le 6\delta\sigma_1 r$$

so by Claim F.2 we must have

$$||v - \overline{v}||_2 \le 6\delta\sigma_1 rn$$

Thus

$$||T' - \overline{T}||_2 = ||M(v' - \overline{v})||_2 \le 6\sigma_1\delta rn$$

and combining with (6) we immediately get the desired. $\qquad\square$

## F.2  Decomposition Step

Now we analyze the decomposition step where we decompose $T'$ into rank-1 components. First, we formally state JENNRICH'S ALGORITHM and its guarantees.

### F.2.1  Tensor Decomposition via Jennrich's Algorithm

JENNRICH'S ALGORITHM is an algorithm for decomposing a tensor, say $T = \sum_{i=1}^{r}(x_i \otimes y_i \otimes z_i)$, into its rank-1 components that works when the fibers of the rank 1 components i.e. $x_1, \ldots, x_r$ are linearly independent (and similar for $y_1, \ldots, y_r$ and $z_1, \ldots, z_r$).

---

**Algorithm 5** JENNRICH'S ALGORITHM

---

**Input**: Tensor $T' \in \mathbb{R}^{n \times n \times n}$ where

$$T' = T + E$$

for some rank-$r$ tensor $T$ and error $E$

Choose unit vectors $a, b \in \mathbb{R}^n$ uniformly at random
Let $T^{(a)}, T^{(b)}$ be $n \times n$ matrices defined as

$$T_{ij}^{(a)} = T'_{i,j,\cdot} \cdot a$$
$$T_{ij}^{(b)} = T'_{i,j,\cdot} \cdot b$$

Let $T_r^{(a)}, T_r^{(b)}$ be obtained by taking the top $r$ principal components of $T^{(a)}, T^{(b)}$ respectively.

Compute the eigendecompositions of $U = T_r^{(a)}(T_r^{(b)})^+$ and $V = \left((T_r^{(a)})^+ T_r^{(b)}\right)^T$ (where for a matrix $M$, $M^+$ denotes the pseudoinverse)
Let $u_1, \ldots, u_r, v_1, \ldots, v_r$ be the eigenvectors computed in the previous step.
Permute the $v_i$ so that for each pair $(u_i, v_i)$, the corresponding eigenvalues are (approximately) reciprocals.
Solve the following for the vectors $w_i$

$$\arg\min \left\|T' - \sum_{i=1}^{r} u_i \otimes v_i \otimes w_i\right\|_2^2$$

Output the rank-1 components $\{u_i \otimes v_i \otimes w_i\}_{i=1}^{r}$

---

Moitra [21] gives a complete analysis of JENNRICH'S ALGORITHM. The result that we need is that as the error $E$ goes to $0$ at an inverse-polynomial rate, JENNRICH'S ALGORITHM recovers the individual rank-1 components to within any desired inverse-polynomial accuracy. Note that the exact polynomial dependencies do not matter for our purposes as they only result in a constant factor change in the number of iterations of alternating minimization that we need to perform.

**Theorem F.3** ([21]). *Let*

$$T = \sum_{i=1}^{r} \sigma_i (x_i \otimes y_i \otimes z_i)$$

*where the $x_i, y_i, z_i$ are unit vectors and $\sigma_1 \geq \cdots \geq \sigma_r > 0$. Assume that the smallest singular value of the matrix with columns given by $x_1, \ldots, x_r$ is at least $c$ and similar for the $y_i$ and $z_i$. Then for any constant $d$, there exists a polynomial $P$ such that if*

$$\|E\|_2 \leq \frac{\sigma_1}{P(n, \frac{1}{c}, \frac{\sigma_1}{\sigma_r})}$$

*then with $1 - \frac{1}{(10n)^d}$ probability, there is a permutation $\pi$ such that the outputs of JENNRICH'S ALGORITHM satisfy*

$$\left\| \sigma_{\pi(i)} (x_{\pi(i)} \otimes y_{\pi(i)} \otimes z_{\pi(i)}) - u_i \otimes v_i \otimes w_i \right\|_2 \leq \sigma_1 \left( \frac{\sigma_r c}{10 \sigma_1 n} \right)^d$$

*for all $1 \leq i \leq r$.*

**Remark.** *Note that the extra factors of $\sigma_1$ in the theorem above are simply to deal with the scaling of the tensor $T$.*

### F.2.2 Uniqueness of Decomposition

In Section F.1, we showed that our estimate $T'$ is close to $T$. We will now show that the components $\widehat{\sigma}_i, \widehat{x}, \widehat{y}, \widehat{z}$ that we obtain by decomposing $T'$ are close to the true components.

**Theorem F.4.** *Consider running the* POST-PROCESSING VIA CONVEX OPTIMIZATION *algorithm with parameter*

$$p_3 = 2 \frac{\log^2 n}{n^2} \left( \frac{10 r \mu}{c} \cdot \frac{\sigma_1}{\sigma_r} \right)^{10}$$

*and input subspaces that satisfy*

$$\max \left( \alpha(V_x, \widehat{V_x}), \alpha(V_y, \widehat{V_y}), \alpha(V_z, \widehat{V_z}) \right) \leq \left( \frac{\sigma_r c}{10 \sigma_1 n} \right)^{10^2}.$$

*With probability at least $0.95$, there exists a permutation $\pi : [n] \to [n]$ and $\epsilon_x, \epsilon_y, \epsilon_z \in \{-1, 1\}$ such that the estimates $\widehat{\sigma}_i, \widehat{x}_i, \widehat{y}_i, \widehat{z}_i$ computed in the* POST-PROCESSING VIA CONVEX OPTIMIZATION *algorithm satisfy*

$$\frac{|\sigma_i - \widehat{\sigma_{\pi(i)}}|}{\sigma_1}, \|x_i - \epsilon_x \widehat{x_{\pi(i)}}\|, \|y_i - \epsilon_y \widehat{y_{\pi(i)}}\|, \|z_i - \epsilon_z \widehat{z_{\pi(i)}}\| \leq \epsilon$$

*where $\epsilon = \left( \frac{c \sigma_r}{10 n \sigma_1} \right)^{20}$*

*Proof.* By Lemma F.1 we have with probability at least $0.98$ that

$$\|T - T'\|_2 \leq \sigma_1 \left( \frac{\sigma_r c}{10 \sigma_1 n} \right)^{99}$$

Now by the robust analysis of JENNRICH'S ALGORITHM (see Theorem F.3) we know that with $0.97$ probability, for the components $T_1, \ldots, T_r$ that we obtain in the decomposition of $T'$, there is a permutation $\pi$ such that

$$\|T_{\pi(i)} - \sigma_i x_i \otimes y_i \otimes z_i\|_2 \leq \sigma_1 \left( \frac{\sigma_r c}{10 n \sigma_1} \right)^{40}$$

for all $1 \leq i \leq r$. We write $T_{\pi(i)} = \widehat{\sigma_{\pi(i)}} \widehat{x_{\pi(i)}} \otimes \widehat{y_{\pi(i)}} \otimes \widehat{z_{\pi(i)}}$ where $\widehat{x_{\pi(i)}}, \widehat{y_{\pi(i)}}, \widehat{z_{\pi(i)}}$ are unit vectors and $\widehat{\sigma_{\pi(i)}}$ is nonnegative. Note that this decomposition is clearly unique up to flipping the signs on the unit vectors. Then

$$||\widehat{\sigma_{\pi(i)}} \widehat{x_{\pi(i)}} \otimes \widehat{y_{\pi(i)}} \otimes \widehat{z_{\pi(i)}} - \sigma_i x_i \otimes y_i \otimes z_i||_2 \geq \left| \left\| \widehat{\sigma_{\pi(i)}} \widehat{x_{\pi(i)}} \otimes \widehat{y_{\pi(i)}} \otimes \widehat{z_{\pi(i)}} \right\|_2 - ||\sigma_i x_i \otimes y_i \otimes z_i||_2 \right| = |\sigma_i - \widehat{\sigma_{\pi(i)}}|$$

Thus

$$|\sigma_i - \widehat{\sigma_{\pi(i)}}| \leq \sigma_1 \left( \frac{\sigma_r c}{10 n \sigma_1} \right)^{40}$$

Now let $x_\perp$ be the projection of $\widehat{x_{\pi(i)}}$ onto the orthogonal complement of $x_i$. WLOG $\widehat{x_{\pi(i)}} \cdot x_i \geq 0$. Otherwise we can set $\epsilon_x = -1$. Then

$$x_\perp \geq 0.5 ||x_i - \widehat{x_{\pi(i)}}||_2$$

Also

$$||\widehat{\sigma_{\pi(i)}} \widehat{x_{\pi(i)}} \otimes \widehat{y_{\pi(i)}} \otimes \widehat{z_{\pi(i)}} - \sigma_i x_i \otimes y_i \otimes z_i||_2 \geq ||\widehat{\sigma_{\pi(i)}} x_\perp \otimes \widehat{y_{\pi(i)}} \otimes \widehat{z_{\pi(i)}}||_2 \geq 0.1 \sigma_r ||x_i - \widehat{x_{\pi(i)}}||_2$$

Thus

$$||x_i - \widehat{x_{\pi(i)}}||_2 \leq \left( \frac{\sigma_r c}{10 n \sigma_1} \right)^{20}$$

and similar for $\widehat{y_{\pi(i)}}, \widehat{z_{\pi(i)}}$, completing the proof. $\qquad\square$

From now on we will assume $\pi$ is the identity permutation and $\epsilon_x = \epsilon_y = \epsilon_z = 1$. It is clear that these assumptions are without loss of generality. We now have

**Assertion F.5.** *For all $1 \leq i \leq r$*

$$\frac{|\sigma_i - \widehat{\sigma}_i|}{\sigma_1}, ||x_i - \widehat{x}_i||, ||y_i - \widehat{y}_i||, ||z_i - \widehat{z}_i|| \leq \epsilon$$

*where $\epsilon = \left( \frac{c \sigma_r}{10 n \sigma_1} \right)^{20}$.*

## G   Exact Completion via Convex Optimization

In the last step of our algorithm, once we have estimates $\widehat{\sigma}_i, \widehat{x}_i, \widehat{y}_i, \widehat{z}_i$, we solve the following optimization problem which we claim is strongly convex. Let $S_\sim$ be the set of observed entries in $T_\sim$. For each $1 \leq i \leq r$ let $y_i'$ be the unit vector in $\mathsf{span}(\widehat{y}_1, \dots, \widehat{y}_r)$ that is orthogonal to $\widehat{y}_1, \dots, \widehat{y}_{i-1}, \widehat{y}_{i+1}, \dots, \widehat{y}_r$. Define $z_i'$ similarly. We solve

$$\min_{a_i, b_i, c_i} \left\| \left( T - \sum_{i=1}^r (\widehat{\sigma}_i(\widehat{x}_i + a_i)) \otimes (\widehat{y}_i + b_i) \otimes (\widehat{z}_i + c_i) \right) \Bigg|_{S_\sim} \right\|_2^2 \qquad (7)$$

with the constraints

- $0 \leq ||a_i||_\infty, ||b_i||_\infty, ||c_i||_\infty \leq \left( \frac{c \sigma_r}{10 n \sigma_1} \right)^{10}$ for all $1 \leq i \leq r$.
- $b_i \cdot y_i' = 0$ and $c_i \cdot z_i' = 0$ for all $1 \leq i \leq r$.

Assume that Assertion F.5 holds. We will prove the following three lemmas which will imply that the optimization problem can be solved efficiently and yields the desired solution.

**Lemma G.1.** *Assuming that Assertion F.5 holds, an optimal solution of (7) is*

$$a_i = \frac{\sigma_i}{\widehat{\sigma}_i} \frac{(y_i \cdot y_i')(z_i \cdot z_i')}{(\widehat{y}_i \cdot y_i')(\widehat{z}_i \cdot z_i')} x_i - \widehat{x}_i$$

$$b_i = \frac{\widehat{y}_i \cdot y_i'}{y_i \cdot y_i'} y_i - \widehat{y}_i$$

$$c_i = \frac{\widehat{z}_i \cdot z_i'}{z_i \cdot z_i'} z_i - \widehat{z}_i$$

**Lemma G.2.** *Assuming that Assertion F.5 holds, with $1 - \frac{1}{n^{10}}$ probability over the random sample $S_\sim$, the objective function in (7) is $\frac{\sigma_r^2 c^6}{10r} \cdot \frac{\log^2 n}{n^2} \left( \frac{10r\mu}{c} \cdot \frac{\sigma_1}{\sigma_r} \right)^{10}$-strongly convex.*

**Lemma G.3.** *Assuming that Assertion F.5 holds, with $1 - \frac{1}{n^{10}}$ probability over the random sample $S_\sim$, the objective function in (7) is $20\sigma_1^2 r \cdot \frac{\log^2 n}{n^2} \left( \frac{10r\mu}{c} \cdot \frac{\sigma_1}{\sigma_r} \right)^{10}$-smooth.*

First we demonstrate why these lemmas are enough to finish the proof of Theorem 3.2.

*Proof of Theorem 3.2.* Note that for the solution stated in Lemma G.1, the value of the objective in (7) is 0 and thus the solution is a local minimum. Lemma G.2 implies that the optimization problem is strongly convex and thus this is actually the global minimum. Also since the ratio of the strong convexity and smoothness parameters is $\frac{200\sigma_1^2 r^2}{\sigma_r^2 c^6}$, the optimization can be solved efficiently (see [23]). For the solution in Lemma G.1, the output of our FULL EXACT TENSOR COMPLETION ALGORITHM is exactly $T$. Thus combining Theorem B.1, Corollary E.9, and Theorem F.4 with Lemma G.1, Lemma G.2 and Lemma G.3, we are done. $\qquad\square$

## G.1 The True Solution Satisfies the Constraints

First we show that the true solution $T$ can be recovered while satisfying the constraints.

*Proof of Lemma G.1.* When

$$a_i = \frac{\sigma_i}{\widehat{\sigma}_i} \frac{(y_i \cdot y_i')(z_i \cdot z_i')}{(\widehat{y}_i \cdot y_i')(\widehat{z}_i \cdot z_i')} x_i - \widehat{x}_i$$

$$b_i = \frac{\widehat{y}_i \cdot y_i'}{y_i \cdot y_i'} y_i - \widehat{y}_i$$

$$c_i = \frac{\widehat{z}_i \cdot z_i'}{z_i \cdot z_i'} z_i - \widehat{z}_i$$

then the value of the objective is 0 and we exactly recover $T$. It remains to show that this solution satisfies the constraints. It is immediate that the second constraint is satisfied. We now verify that the first constraint is also satisfied. Note that the smallest singular value of $V_y$ is at least $c$. Thus the smallest singular value of the matrix with columns $\widehat{y}_1, \ldots, \widehat{y}_r$ is at least $c - \epsilon r$. In particular $\widehat{y}_i \cdot y_i'$ must be at least $c - \epsilon r$. Also the difference between $\widehat{y}_i \cdot y_i'$ and $y_i \cdot y_i'$ is at most $\epsilon$. Thus

$$1 - \frac{2\epsilon r}{c} \leq \frac{\widehat{y}_i \cdot y_i'}{y_i \cdot y_i'} \leq 1 + \frac{2\epsilon r}{c}$$

Combining this with the fact that $\frac{|\sigma_i - \widehat{\sigma}_i|}{\sigma_1}, \|x_i - \widehat{x}_i\|, \|y_i - \widehat{y}_i\|, \|z_i - \widehat{z}_i\| \leq \epsilon$, it is clear that the first constraint is satisfied. $\qquad\square$

## G.2 The Optimization Problem is Strongly Convex

To show that the optimization problem is strongly convex, we will compute the Hessian of the objective function. Let $m$ be the magnitude of the largest entry of

$$T - \sum_{i=1}^r \widehat{\sigma}_i \widehat{x}_i \otimes \widehat{y}_i \otimes \widehat{z}_i$$

Note $m \leq 10r\sigma_1\epsilon$ where $\epsilon = \left( \frac{c\sigma_r}{10n\sigma_1} \right)^{20}$.

Next, let $\widehat{\sigma} = \max(\widehat{\sigma}_1, \ldots, \widehat{\sigma}_r)$. Note $\widehat{\sigma} \leq 2\sigma_1$. Also define

$$D = \sum_{i=1}^r \widehat{\sigma}_i a_i \otimes \widehat{y}_i \otimes \widehat{z}_i + \widehat{\sigma}_i \widehat{x}_i \otimes b_i \otimes \widehat{z}_i + \widehat{\sigma}_i \widehat{x}_i \otimes \widehat{y}_i \otimes c_i$$

Note that the objective function can be written as

$$||D|_{S_\sim}||_2^2 + P(a_i, b_i, c_i) \tag{8}$$

where $P$ is a polynomial with the following property: all terms of $P$ of degree 2 have coefficients with magnitude at most $(10nr)^6 m\widehat{\sigma}$ and all coefficients for higher degree terms have magnitude at most $(10nr)^6 \widehat{\sigma}^2$. Now to prove strong convexity we will lower bound the smallest singular value of the Hessian of $||D|_{S_\sim}||_2^2$ (with respect to the variables $a_i, b_i, c_i$). Since $m, a_i, b_i, c_i$ are all small, we can ensure that the contribution of $P$ does not affect the strong convexity and this will complete the proof.

### G.2.1 Understanding the Hessian when $S_\sim$ contains all entries

First we consider $H_0$, the Hessian of $||D||_2^2$ i.e. when we are not restricted to the set of entries in $S_\sim$.

**Claim G.4.** *The smallest eigenvalue of $H_0$ is at least* $\frac{\sigma_r^2 c^6}{2r}$

*Proof.* Consider a directional vector $v = (\Delta_{a_1}, \Delta_{b_1}, \Delta_{c_1}, \ldots, \Delta_{a_r}, \Delta_{b_r}, \Delta_{c_r})$. Then

$$||v^T H_0 v||_2^2 = \left\| \sum_{i=1}^{r} (\widehat{\sigma}_i \Delta_{a_i} \otimes \widehat{y_i} \otimes \widehat{z_i} + \widehat{\sigma}_i \widehat{x_i} \otimes \Delta_{b_i} \otimes \widehat{z_i} + \widehat{\sigma}_i \widehat{x_i} \otimes \widehat{y_i} \otimes \Delta_{c_i}) \right\|_2^2$$

We now lower bound the RHS. For each $i$, let $\Delta_{a_i}^-$ be the projection of $\Delta_{a_i}$ onto $\mathsf{span}(\widehat{x_1}, \ldots, \widehat{x_r})$ and let $\Delta_{a_i}^\perp$ be the projection of $\Delta_{a_i}$ onto the orthogonal complement of $\mathsf{span}(\widehat{x_1}, \ldots, \widehat{x_r})$. Define $\Delta_{b_i}^-, \Delta_{b_i}^\perp, \Delta_{c_i}^-, \Delta_{c_i}^\perp$ similarly. We want to lower bound the squared Frobenius norm of

$$\sum_{i=1}^{r} (\widehat{\sigma}_i \Delta_{a_i} \otimes \widehat{y_i} \otimes \widehat{z_i} + \widehat{\sigma}_i \widehat{x_i} \otimes \Delta_{b_i} \otimes \widehat{z_i} + \widehat{\sigma}_i \widehat{x_i} \otimes \widehat{y_i} \otimes \Delta_{c_i}) =$$

$$\sum_{i=1}^{r} (\widehat{\sigma}_i \Delta_{a_i}^- \otimes \widehat{y_i} \otimes \widehat{z_i} + \widehat{\sigma}_i \widehat{x_i} \otimes \Delta_{b_i}^- \otimes \widehat{z_i} + \widehat{\sigma}_i \widehat{x_i} \otimes \widehat{y_i} \otimes \Delta_{c_i}^-)$$

$$+ \sum_{i=1}^{r} \widehat{\sigma}_i \Delta_{a_i}^\perp \otimes \widehat{y_i} \otimes \widehat{z_i} + \sum_{i=1}^{r} \widehat{\sigma}_i \widehat{x_i} \otimes \Delta_{b_i}^\perp \otimes \widehat{z_i} + \sum_{i=1}^{r} \widehat{\sigma}_i \widehat{x_i} \otimes \widehat{y_i} \otimes \Delta_{c_i}^\perp$$

Let the four sums above be $A, B, C, D$ respectively. $A, B, C, D$ are pairwise orthogonal. Thus it suffices to lower bound the Frobenius norm of each of them individually. First we lower bound the Frobenius norm of $A$. Since $\Delta_{a_i}^-$ is in $\mathsf{span}(\widehat{x_1}, \ldots, \widehat{x_r})$, it can be written as a linear combination of $\widehat{x_1}, \ldots, \widehat{x_r}$ say

$$\Delta_{a_i}^- = a_i^{(1)} \widehat{x_1} + \cdots + a_i^{(r)} \widehat{x_r}$$

Furthermore

$$\left(a_i^{(1)}\right)^2 + \cdots + \left(a_i^{(r)}\right)^2 \geq \frac{||\Delta_{a_i}^-||_2^2}{r}$$

We can use the same argument for $\Delta_{b_i}^-, \Delta_{c_i}^-$. Also, since in our optimization problem we have the constraints $b_i \cdot y_i' = 0, c_i \cdot z_i' = 0$, we know that the coefficients $b_i^{(i)}, c_i^{(i)}$ are 0. Thus we can write $A$ as a sum

$$A = \sum_{i=1}^{r} \sum_{j=1}^{r} \widehat{\sigma}_i a_i^{(j)} \widehat{x_j} \otimes \widehat{y_i} \otimes \widehat{z_i} + \sum_{i=1}^{r} \sum_{j=1, j \neq i}^{r} \widehat{\sigma}_i \widehat{x_i} \otimes b_i^{(j)} \widehat{y_j} \otimes \widehat{z_i} + \sum_{i=1}^{r} \sum_{j=1, j \neq i}^{r} \widehat{\sigma}_i \widehat{x_i} \otimes \widehat{y_i} \otimes c_i^{(j)} \widehat{z_j}$$

Note that the above is a linear combination of terms of the form $\widehat{x_i} \otimes \widehat{y_j} \otimes \widehat{z_k}$ and each term appears at most once. Furthermore, the sum of the squares of the coefficients is at least

$$\sum_{i=1}^{r} \min(\widehat{\sigma_1}, \ldots, \widehat{\sigma_r})^2 \left( \frac{||\Delta_{a_i}^-||_2^2}{r} + \frac{||\Delta_{b_i}^-||_2^2}{r} + \frac{||\Delta_{c_i}^-||_2^2}{r} \right)$$

Next, observe that the smallest singular value of the matrix with columns given by $\widehat{x_i} \otimes \widehat{y_j} \otimes \widehat{z_k}$ for $1 \leq i, j, k \leq r$ is at least $(c - r\epsilon)^3$. Thus

$$||A||_2^2 \geq (c - r\epsilon)^6 \sum_{i=1}^{r} \min(\widehat{\sigma_1}, \ldots, \widehat{\sigma_r})^2 \left( \frac{||\Delta_{a_i}^-||_2^2}{r} + \frac{||\Delta_{b_i}^-||_2^2}{r} + \frac{||\Delta_{c_i}^-||_2^2}{r} \right)$$

Now we lower bound the squared Frobenius norm of $B$. Each slice of $B$ is a linear combination of $\widehat{y}_1 \otimes \widehat{z}_1, \ldots, \widehat{y}_r \otimes \widehat{z}_r$. Note the matrix with columns given by $\widehat{y}_i \otimes \widehat{z}_j$ for $1 \leq i, j \leq r$ has smallest singular value at least $(c - r\epsilon)^2$. Thus if we let $\Delta_{a_i}^{\perp[j]}$ be the $j^{\text{th}}$ entry of $\Delta_{a_i}^{\perp}$ then the sum of the squares of the entries in the $j^{\text{th}}$ layer of $B$ is at least

$$\min(\widehat{\sigma_1}, \ldots, \widehat{\sigma_r})^2 (c - r\epsilon)^4 \left( \left( \Delta_{a_1}^{\perp[j]} \right)^2 + \cdots + \left( \Delta_{a_r}^{\perp[j]} \right)^2 \right).$$

Overall we get

$$||B||_2^2 \geq \sum_{j=1}^{n} \min(\widehat{\sigma_1}, \ldots, \widehat{\sigma_r})^2 (c - r\epsilon)^4 \left( \left( \Delta_{a_1}^{\perp[j]} \right)^2 + \cdots + \left( \Delta_{a_r}^{\perp[j]} \right)^2 \right)$$

$$\geq \min(\widehat{\sigma_1}, \ldots, \widehat{\sigma_r})^2 (c - r\epsilon)^4 \sum_{i=1}^{r} ||\Delta_{a_i}^{\perp}||_2^2.$$

Similarly

$$||C||_2^2 \geq \min(\widehat{\sigma_1}, \ldots, \widehat{\sigma_r})^2 (c - r\epsilon)^4 \sum_{i=1}^{r} ||\Delta_{b_i}^{\perp}||_2^2$$

$$||D||_2^2 \geq \min(\widehat{\sigma_1}, \ldots, \widehat{\sigma_r})^2 (c - r\epsilon)^4 \sum_{i=1}^{r} ||\Delta_{c_i}^{\perp}||_2^2.$$

Overall we have

$$||v^T H_0 v||_2^2 = ||A||_2^2 + ||B||_2^2 + ||C||_2^2 + ||D||_2^2 \geq \frac{\min(\widehat{\sigma_1}, \ldots, \widehat{\sigma_r})^2 (c - r\epsilon)^6}{r} ||v||_2^2 \geq \frac{\sigma_r^2 c^6}{2r} ||v||_2^2$$

and we get that the smallest eigenvalue of $H_0$ is at least $\frac{\sigma_r^2 c^6}{2r}$. $\qquad\qquad\square$

### G.2.2 Understanding the Hessian when $|S_\sim|$ is small

To prove Lemma G.2, we want to go from a bound on the Hessian of $||D||_2^2$ to a bound on the Hessian of $||D|_{S_\sim}||_2^2$. We will then use the fact that the Hessian of $P(a_i, b_i, c_i)$ is small and cannot substantially affect the strong convexity.

*Proof of Lemma G.2.* Note that $||D||_2^2$ is a sum of $n^3$ terms each of which is the square of a linear function (corresponding to an entry). Each of these terms contributes a rank-1 term to the Hessian. Furthermore, since all entries of $\widehat{x}_i, \widehat{y}_i, \widehat{z}_i$ are at most $\sqrt{\frac{\mu r}{n}} + \epsilon$, the operator norm of each of these rank 1 terms is at most $9\widehat{\sigma}^2 r^2 \left( \sqrt{\frac{\mu r}{n}} + \epsilon \right)^4 \leq \frac{10 r^4 \mu^2 \sigma_1^2}{n^2}$.

If we add each entry to $S_\sim$ with probability $p = \frac{\log^2 n}{n^2} \left( \frac{10 r \mu}{c} \cdot \frac{\sigma_1}{\sigma_r} \right)^{10}$ then by Claim C.9, with at least $1 - \frac{1}{n^{10}}$ probability, the sum of the rank 1 terms corresponding to entries of $S$ has smallest singular value at least $\frac{\sigma_r^2 c^6}{4r} p$.

We have shown that with high probability, the smallest eigenvalue of the Hessian of $||D|_{S_\sim}||_2^2$ is at least $\frac{\sigma_r^2 c^6}{4r} p$. It remains to note that the Hessian of $P(a_i, b_i, c_i)$ has operator norm at most $\frac{\sigma_r^2 c^6}{10 r n^2}$ for all $a_i, b_i, c_i$ in the feasible set and thus the optimization problem we formulated is strongly convex with parameter

$$\frac{\sigma_r^2 c^6}{10 r} p = \frac{\sigma_r^2 c^6}{10 r} \cdot \frac{\log^2 n}{n^2} \left( \frac{10 r \mu}{c} \cdot \frac{\sigma_1}{\sigma_r} \right)^{10}$$

$\square$

### G.3 The Optimization Problem is Smooth

The proof that the objective function is smooth will follow a similar approach to that in Section G.2. We use the same notation as the previous section. Again, the first step will be to consider the Hessian $H_0$ of $||D||_2^2$ when we are not restricted to the set of entries in $S_\sim$.

**Claim G.5.** *The largest eigenvalue of $H_0$ is at most $5\sigma_1^2 r$.*

*Proof.* Consider a directional vector $v = (\Delta_{a_1}, \Delta_{b_1}, \Delta_{c_1}, \dots, \Delta_{a_r}, \Delta_{b_r}, \Delta_{c_r})$. Then

$$||v^T H_0 v||_2^2 = \left\| \sum_{i=1}^r (\widehat{\sigma}_i \Delta_{a_i} \otimes \widehat{y}_i \otimes \widehat{z}_i + \widehat{\sigma}_i \widehat{x}_i \otimes \Delta_{b_i} \otimes \widehat{z}_i + \widehat{\sigma}_i \widehat{x}_i \otimes \widehat{y}_i \otimes \Delta_{c_i}) \right\|_2^2.$$

Thus,

$$||v^T H_0 v||_2^2 \leq \left( \sum_{i=1}^r \|\widehat{\sigma}_i \Delta_{a_i} \otimes \widehat{y}_i \otimes \widehat{z}_i\|_2 + \|\widehat{\sigma}_i \widehat{x}_i \otimes \Delta_{b_i} \otimes \widehat{z}_i\|_2 + \|\widehat{\sigma}_i \widehat{x}_i \otimes \widehat{y}_i \otimes \Delta_{c_i}\|_2 \right)^2$$

$$\leq \left( \widehat{\sigma} \sum_{i=1}^r (\|\Delta_{a_i}\|_2 + \|\Delta_{b_i}\|_2 + \|\Delta_{c_i}\|_2) \right)^2$$

$$\leq 5\sigma_1^2 r \left( \sum_{i=1}^r \|\Delta_{a_i}\|_2^2 + \|\Delta_{b_i}\|_2^2 + \|\Delta_{c_i}\|_2^2 \right)$$

$$= 5\sigma_1^2 r \|v\|_2^2$$

which immediately implies the desired.

$\square$

Now we can complete the proof of Lemma G.3 in the same way we proved Lemma G.2 through a matrix Chernoff bound and the fact that the Hessian of $P(a_i, b_i, c_i)$ is small.

*Proof of Lemma G.3.* Note that $||D_{S_\sim}||_2^2$ is a sum of $|S_\sim|$ terms each of which is the square of a linear function (corresponding to an entry). Each of these terms contributes a rank-1 term to the Hessian. Furthermore, since all entries of $\widehat{x}_i, \widehat{y}_i, \widehat{z}_i$ are at most $\sqrt{\frac{\mu r}{n}} + \epsilon$, the operator norm of each of these rank 1 terms is at most $9\widehat{\sigma}^2 r^2 \left( \sqrt{\frac{\mu r}{n}} + \epsilon \right)^4 \leq \frac{10 r^4 \mu^2 \sigma_1^2}{n^2}$.

If we add each entry to $S_\sim$ with probability $p = \frac{\log^2 n}{n^2} (\frac{10 r \mu}{c} \cdot \frac{\sigma_1}{\sigma_r})^{10}$ then by Claim C.10, with at least $1 - \frac{1}{n^{10}}$ probability, the sum of the rank 1 terms corresponding to entries of $S$ has largest singular value at most $10\sigma_1^2 r p$.

We have shown that with high probability, the largest eigenvalue of the Hessian of $||D|_{S_\sim}||_2^2$ is at most $10\sigma_1^2 r p$. It remains to note that the Hessian of $P(a_i, b_i, c_i)$ has operator norm at most $\frac{\sigma_r^2 c^6}{10 r n^2}$ for all $a_i, b_i, c_i$ in the feasible set and thus the optimization problem we formulated is smooth with parameter

$$20\sigma_1^2 r p = 20\sigma_1^2 r \cdot \frac{\log^2 n}{n^2} \left( \frac{10 r \mu}{c} \cdot \frac{\sigma_1}{\sigma_r} \right)^{10}$$

$\square$

# H    Nearly Linear Time Implementation

Now we show how to implement our FULL EXACT TENSOR COMPLETION algorithm with running time that is essentially linear in the number of observations (up to $\text{poly}(r, \log n, \sigma_1/\sigma_r, \mu, 1/c)$-factors). We will assume that our observations are in a list of tuples giving the coordinates and value i.e. $(i, j, k, T_{ijk})$. Throughout this section, we use $\zeta$ to denote a quantity that is $\text{poly}(r, \log n, \sigma_1/\sigma_r, \mu, 1/c)$.

## H.1    Initialization

We will construct $\widehat{B}$ implicitly, i.e. we will store the coordinates of all of its nonzero entries and their values. To do this we can enumerate over all pairs $(j, k) \in [n]^2$ such that there is some $i \in [n]$ for

which we observe $T_{ijk}$. For each of these pairs $(j,k)$ we take all pairs $i, i'$ such that $T_{ijk}$ and $T_{i'jk}$ are observed (we may have $i = i'$) and update $\widehat{B}_{ii'}$. For each pair $(j,k)$, let $X_{j,k}$ be the number of distinct $i$ for which we observe $T_{ijk}$. Note

$$\mathbb{E}\left[\sum_{j=1}^{n}\sum_{k=1}^{n} X_{j,k}^2\right] \le \zeta n^{3/2}$$

so the time complexity of this step and the sparsity of $\widehat{B}$ is essentially linear in the number of observations.

Next to compute the top-$r$ singular vectors of $\widehat{B}$ we can use matrix powering (with the implicit sparse representation for $\widehat{B}$). Note Lemma D.1 implies that there is a sufficient gap between the $r^{\text{th}}$ and $r + 1^{\text{st}}$ singular values of $\widehat{B}$ that matrix powering converges within $\zeta$ rounds. It is clear that the remainder of the steps in the initialization algorithm can be completed in nearly linear time.

## H.2 Alternating Minimization

Note that for the least squares optimization problem, it suffices to solve the optimization for each row separately. For the rows of $U_x(\widehat{T_{t+1}})$, let $o_1, \ldots, o_n$ be the number of observations in each row. The least squares problems for the rows have sizes $o_1, \ldots, o_n$ respectively. Instead of constructing the full matrix $B_t$ (which has size $n^2$), we only need to compute the columns of $B_t$ that correspond to actual observations, which can be done using the matrices $V_x^t, V_y^t, V_z^t$. Thus, the least squares problems can be solved in time essentially linear in $o_1 + \cdots + o_n$. Overall, this implies that all of the alternating minimization steps can be completed in nearly linear time.

## H.3 Post-Processing

Note the projection step can be solved in nearly linear time and from it we obtain a representation of $T'$ as a sum of $r^3$ rank-1 tensors (corresponding to the basis given by $\widehat{V_x} \otimes \widehat{V_y} \otimes \widehat{V_z}$.

### H.3.1 Jennrich's Algorithm

Note we have an implicit representation of the tensor $T'$ that we are decomposing as a sum of $r^3$ rank-1 components. Thus, we can compute implicit representations of $T^{(a)}, T^{(b)}$ each as a sum of $r^3$ rank-1 matrices. Next, we can use matrix powering with the implicit representations to compute the top $r$ principal components for $T^{(a)}, T^{(b)}$ (note the analysis in [21] implies there is a sufficient gap between the $r^{\text{th}}$ and $r + 1^{\text{st}}$ singular values of these matrices). Now, we can compute the pseudo-inverses of the rank-$r$ matrices $T_r^{(a)}, T_r^{(b)}$ (written implicitly as the sum of $r$ rank-1 matrices) in $n\text{poly}(r)$ operations.

We can compute the eigendecompositions of $U = T_r^{(a)}(T_r^{(b)})^{+}$ and $V = \left((T_r^{(a)})^{+}T_r^{(b)}\right)^{T}$ using implicit matrix powering again (the analysis in [21] implies that with $0.99$ probability, the eigenvalues of these matrices are sufficiently separated). These operations all take $n\zeta$ time. Finally, we show that once we have $(u_1, v_1), \ldots, (u_r, v_r)$, we can solve for $w_1, \ldots, w_r$. To do this, instead of solving the full least squares problem, we will choose a random subset of $\text{poly}(r, \log n, \sigma_1/\sigma_r, \mu, 1/c)$ entries within each layer of the tensor $T'$ and solve the least squares optimization restricted to those entries.

To see why this works, first note that the subspaces spanned by $u_1, \ldots, u_r$ and $v_1, \ldots, v_r$ are $2\mu$-incoherent (the proof of Theorem F.4 implies that $u_1, \ldots, u_r$ and $v_1, \ldots, v_r$ are close to the true factors up to some permutation). Next, let $A$ be the matrix with columns given by $u_1 \otimes v_1, \ldots, u_r \otimes v_r$. Note that if $A'$ is a matrix constructed by selecting a random subset of $\text{poly}(r, \log n, \sigma_1/\sigma_r, \mu, 1/c)$ rows of $A$, then with $1 - \frac{1}{n^{10}}$ probability, $A'$ is well-conditioned (by incoherence and the matrix Chernoff bound in Claim C.9). Since $A'$ is well-conditioned, the solution to the restricted least squares optimization problem must still be close to the true solution.

Thus, the entire least-squares optimization can be completed in $n\zeta$ time. Overall, we conclude that the tensor decomposition step can be completed in $n\zeta$ time.

### H.3.2 Convex Optimization

It remains to show that the final optimization problem can be solved in nearly linear time. Note the size of the optimization problem is $n\mathrm{poly}(r, \log n, \sigma_1/\sigma_r, \mu, 1/c)$. Lemma G.2 and Lemma G.3 imply that the condition number of this convex optimization problem is $\mathrm{poly}(r, \log n, \sigma_1/\sigma_r, \mu, 1/c)$ so it can be solved in $n\zeta$ time.

# I   Code for Experiments

```
import numpy as np
from sklearn.decomposition import TruncatedSVD
from sklearn.linear_model import LinearRegression

#size of tensor
n = 200
#CP rank of tensor
r = 4
#number of observations
num_samples = 50000
p = min(float(num_samples/ (n*n*n)), 1)

#Whether initialization is random or computed using initialization step
randominit = True

#Whether the underlying tensor will have correlated components
correlated = True

#Whether observations are exact or noisy
noisy = True
noise_size = 0.1

#which algorithm to run
#can be "Matrix Alt Min, Tensor Powering, Subspace Powering, all"
which_alg = "Tensor␣Powering"

save_file = "tensorpowering+noisy200_4_50000.csv"

#Number of different tensors to run on
num_runs = 100

#Number of iterations per run
num_iter = 400

#min error threshold
threshold = 10**(-13)

#generate random uncorrelated tensor
def gen(n,r):
        coeffs = np.ones(r)
        x_vecs = np.random.normal(0,1,(r,n))
        y_vecs = np.random.normal(0,1,(r,n))
        z_vecs = np.random.normal(0,1,(r,n))
        return (coeffs, x_vecs,y_vecs,z_vecs)
```

```
#generate random correlated tensor
def gen_biased(n,r):
        coeffs = np.zeros(r)
        x_vecs = np.zeros((r,n))
        y_vecs = np.zeros((r,n))
        z_vecs = np.zeros((r,n))
        for i in range(r):
                coeffs[i] = 0.5**i
                if(i==0):
                        x_vecs[i] = np.sqrt(n) *
                        normalize(np.random.normal(0,1,n))
                        y_vecs[i] = np.sqrt(n) *
                        normalize(np.random.normal(0,1,n))
                        z_vecs[i] = np.sqrt(n) *
                        normalize(np.random.normal(0,1,n))
                else:
                        x_vecs[i] = np.sqrt(n) *
                        normalize(np.random.normal(0,0.5,n) + x_vecs[0])
                        y_vecs[i] = np.sqrt(n) *
                        normalize(np.random.normal(0,0.5,n) + y_vecs[0])
                        z_vecs[i] = np.sqrt(n) *
                        normalize(np.random.normal(0,0.5,n) + z_vecs[0])
        return (coeffs, x_vecs,y_vecs,z_vecs)

#evaluate tensor given coordinates
def T(i,j,k, coeffs, x_vecs, y_vecs, z_vecs):
        ans = 0
        for a in range(r):
                ans += coeffs[a] * x_vecs[a][i] * y_vecs[a][j] * z_vecs[a][k]
        return ans

#sample observations, a is num_samples
#returns 3 lists of coordinates
def sample(a):
        samples = np.random.choice(n**3, a, replace=False)
        x_coords = samples%n
        y_coords = (((samples - x_coords)/n)%n).astype(int)
        z_coords = (((samples - n*y_coords - x_coords)/(n*n))%n).astype(int)
        return (x_coords, y_coords, z_coords)

#Given samples and tensor T, construct dictionary x_dict that stores the
    observations
```

```
def fill(x_coords, y_coords, z_coords, coeffs, x_vecs, y_vecs, z_vecs,
    x_dict):
        num_samples = x_coords.size
        for i in range(num_samples):
                #For x_dict coordinates are in order x,y,z
                if(x_coords[i] in x_dict.keys()):
                        if(y_coords[i] in x_dict[x_coords[i]].keys()):
                                if(z_coords[i] in x_dict[x_coords[i]][y_coords
                                    [i]].keys()):
                                        pass
                                else:
                                        x_dict[x_coords[i]][y_coords[i]][
                                            z_coords[i]] = T(x_coords[i] ,
                                            y_coords[i] , z_coords[i], coeffs,
                                            x_vecs, y_vecs, z_vecs)
                        else:
                                x_dict[x_coords[i]][y_coords[i]] = {}
                                x_dict[x_coords[i]][y_coords[i]][z_coords[i]]
                                    = T(x_coords[i] , y_coords[i] , z_coords[i
                                    ], coeffs, x_vecs, y_vecs, z_vecs)
                else:
                        x_dict[x_coords[i]] = {}
                        x_dict[x_coords[i]][y_coords[i]] = {}
                        x_dict[x_coords[i]][y_coords[i]][z_coords[i]] = T(
                            x_coords[i] , y_coords[i] , z_coords[i], coeffs,
                            x_vecs, y_vecs, z_vecs)

#normalize vector
def normalize(v):
        u = v/np.linalg.norm(v)
        return u

#given rxn array, output orthonormal basis
def orthonormalize(V):
        a = len(V)
        b = len(V[0])
        for i in range(a):
                for j in range(i):
                        V[i] = V[i] - np.dot(V[i],V[j])*V[j]
                V[i] = normalize(V[i])
        return V

#implicit sparse matrix multiplication where M is stored as a dictionary
def mult(M,v):
        u = np.zeros(n)
        for coord1 in M.keys():
                for coord2 in M[coord1].keys():
                        u[coord1] += M[coord1][coord2] * v[coord2]
        return u

#Compute initial subspace estimates
def initialization(x_dict):
        M_x = np.zeros((n,n))
```

```python
        for x in x_dict.keys():
                for y in x_dict[x].keys():
                        for z1 in x_dict[x][y].keys():
                                for z2 in x_dict[x][y].keys():
                                        val = x_dict[x][y][z1] * x_dict[x][y][
                                            z2]
                                        if(z1 == z2):
                                                val = val/p
                                        else:
                                                val = val/(p*p)
                                        M_x[z1][z2] += val
        svd = TruncatedSVD(n_components=r)
        svd.fit(M_x)
        return(svd.components_)

#Unfold and perform matrix completion via altmin
def matrix_altmin(V_x, V_yz):
        #Solve for next iteration of x
        lsq_solution = []
        for i in range(n):
                features = []
                target = []
                for y_coord in x_dict[i].keys():
                        for z_coord in x_dict[i][y_coord].keys():
                                features.append(V_yz[n*y_coord + z_coord])
                                target.append(x_dict[i][y_coord][z_coord])

                features = np.array(features)
                target = np.array(target)

                reg = LinearRegression(fit_intercept = False).fit(features,
                    target)
                lsq_solution.append(reg.coef_)

        x_solution = np.array(lsq_solution)

        #Solve for next iteration of yz
        lsq_solution2 = []
        for i in range(n):
                for j in range(n):
                        features = []
                        target = []
                        if i in y_dict.keys() and j in y_dict[i].keys():
                                for x_coord in y_dict[i][j].keys():
                                        features.append(x_solution[x_coord])
                                        target.append(y_dict[i][j][x_coord])
                                features = np.array(features)
                                target = np.array(target)

                                reg = LinearRegression(fit_intercept = False).
                                    fit(features, target)
                                lsq_solution2.append(reg.coef_)
                        else:
                                lsq_solution2.append(np.zeros(r))

        newV_x = x_solution
```

```python
        newV_yz =np.array(lsq_solution2)
        return(newV_x, newV_yz)

#Normalized MSE for unfolded matrix completion
def eval_error_matrix(V_x,V_yz):
        #take random sample of entries to speed up evaluation
        num_trials = 1000
        total_error = 0
        total_norm = 0
        for i in range(num_trials):
                x = np.random.randint(n)
                y = np.random.randint(n)
                z = np.random.randint(n)

                prediction = 0
                for j in range(r):
                        prediction += V_x[x][j] * V_yz[n * y + z][j]

                true_val = T(x,y,z, coeffs, x_vecs,y_vecs, z_vecs)

                total_norm += np.square(true_val)
                total_error += np.square(prediction - true_val)
        return np.sqrt(total_error/total_norm)

#altmin for naive tensor powering
def power_altmin(V_x, V_y, V_z , x_dict):

        lsq_solution = []
        for i in range(n):
                features = []
                target = []
                for y_coord in x_dict[i].keys():
                        for z_coord in x_dict[i][y_coord].keys():

                                #subsample to speed up and get "unstuck"
                                check = np.random.randint(2)
                                if(check == 0):
                                        features.append(np.multiply(V_y[y_coord
                                            ], V_z[z_coord]))
                                        target.append(x_dict[i][y_coord][
                                            z_coord])

                features = np.array(features)
                target = np.array(target)

                reg = LinearRegression(fit_intercept = False).fit(features,
                    target)
                lsq_solution.append(reg.coef_)

        lsq_solution = np.array(lsq_solution)
        return(lsq_solution)
```

```python
#Normalized MSE for naive tensor powering
def eval_error_direct(V_x,V_y,V_z, x_dict):

        num_trials = 1000
        total_error = 0
        total_norm = 0
        for i in range(num_trials):
                x = np.random.randint(n)
                y = np.random.randint(n)
                z = np.random.randint(n)

                prediction = 0
                for j in range(r):
                        prediction += V_x[x][j] * V_y[y][j] * V_z[z][j]

                true_val = T(x,y,z, coeffs, x_vecs,y_vecs, z_vecs)

                total_norm += np.square(true_val)
                total_error += np.square(prediction - true_val)
        return np.sqrt(total_error/total_norm)

#altmin for our algorithm
def subspace_altmin(V_x, V_y, V_z , x_dict):

        lsq_solution = []
        for i in range(n):
                features = []
                target = []
                for y_coord in x_dict[i].keys():
                        for z_coord in x_dict[i][y_coord].keys():

                                #subsample to speed up and get "unstuck"
                                check = np.random.randint(2)
                                if(check == 0):
                                        features.append(np.tensordot(V_y[
                                            y_coord], V_z[z_coord] , axes = 0).
                                            flatten())
                                        target.append(x_dict[i][y_coord][
                                            z_coord])

                features = np.array(features)
                target = np.array(target)

                reg = LinearRegression(fit_intercept = False).fit(features,
                    target)
                lsq_solution.append(reg.coef_)

        lsq_solution = np.transpose(np.array(lsq_solution))
        svd = TruncatedSVD(n_components=r)
        svd.fit(lsq_solution)

        return(np.transpose(svd.components_))
```

```python
#Normalized MSE for our algorithm
def eval_error_subspace(V_x,V_y,V_z, x_dict):
        features = []
        target = []
        #Find coefficients in V_x x V_y x V_z basis
        for x_coord in x_dict.keys():
                for y_coord in x_dict[x_coord].keys():
                        for z_coord in x_dict[x_coord][y_coord].keys():

                                #speed up by using less entries
                                check = np.random.randint(10)
                                if(check == 0):
                                        target.append(x_dict[x_coord][y_coord][
                                            z_coord])
                                        part = np.tensordot(V_x[x_coord], V_y[
                                            y_coord], axes = 0).flatten()
                                        full = np.tensordot(part, V_z[z_coord],
                                            axes = 0).flatten()
                                        features.append(full)

        features = np.array(features)
        target = np.array(target)
        reg = LinearRegression(fit_intercept = False).fit(features, target)
        solution_coeffs = reg.coef_
        #print(reg.score(features, target))
        #print(solution_coeffs)

        #Evaluate RMS error
        num_trials = 1000
        total_error = 0
        total_norm = 0
        for i in range(num_trials):
                x = np.random.randint(n)
                y = np.random.randint(n)
                z = np.random.randint(n)

                part = np.tensordot(V_x[x], V_y[y], axes = 0).flatten()
                feature = np.tensordot(part, V_z[z], axes = 0).flatten()
                prediction = np.dot(feature, solution_coeffs)

                true_val = T(x,y,z, coeffs, x_vecs,y_vecs, z_vecs)

                total_norm += np.square(true_val)
                total_error += np.square(prediction - true_val)
        return np.sqrt(total_error/total_norm)

#Keep track of errors for all runs
all_errors = []
```

```
for run in range(num_runs):
        #store error over time for this run
        error = []
        curr_error = 1.0

        #Construct random tensor
        if(correlated):
                coeffs, x_vecs,y_vecs,z_vecs = gen_biased(n,r)
        else:
                coeffs, x_vecs,y_vecs,z_vecs = gen(n,r)
        x_coords,y_coords,z_coords = sample(num_samples)

        #x_dict,y_dict, z_dict each stores all observed entries
        #x_dict has coordinates in order x,y,z
        #y_dict has coordinates in order y,z,x
        #z_dict has coordinates in order z,x,y

        x_dict = {}
        y_dict = {}
        z_dict = {}
        fill(x_coords, y_coords, z_coords, coeffs, x_vecs, y_vecs, z_vecs,
            x_dict)
        fill(y_coords, z_coords, x_coords, coeffs, y_vecs, z_vecs, x_vecs,
            y_dict)
        fill(z_coords, x_coords, y_coords, coeffs, z_vecs, x_vecs, y_vecs,
            z_dict)

        #Add Noise
        if(noisy):
                for x_coord in x_dict.keys():
                        for y_coord in x_dict[x_coord].keys():
                                for z_coord in x_dict[x_coord][y_coord].keys():

                                        x_dict[x_coord][y_coord][z_coord] += np.
                                            random.normal(0,noise_size)
                                        y_dict[y_coord][z_coord][x_coord] += np.
                                            random.normal(0,noise_size)
                                        z_dict[z_coord][x_coord][y_coord] += np.
                                            random.normal(0,noise_size)

        #Initialization
        if(randominit):
                V_x = np.random.normal(0,1,(r,n))
                V_y = np.random.normal(0,1,(r,n))
                V_z = np.random.normal(0,1,(r,n))
                V_x = orthonormalize(V_x)
                V_y = orthonormalize(V_y)
                V_z = orthonormalize(V_z)
                V_x = np.transpose(V_x)
                V_y = np.transpose(V_y)
                V_z = np.transpose(V_z)

        else:
```

```python
        V_x = np.transpose(initialization(y_dict))
        V_y = np.transpose(initialization(z_dict))
        V_z = np.transpose(initialization(x_dict))

#For unfolding and matrix completion
V_xmat = np.random.normal(0,1, (r,n))
V_yzmat = np.random.normal(0,1, (r, n*n))
V_xmat = orthonormalize(V_xmat)
V_yzmat = orthonormalize(V_yzmat)
V_xmat = np.transpose(V_xmat)
V_yzmat = np.transpose(V_yzmat)

V_x2 = np.copy(V_x)
V_y2 = np.copy(V_y)
V_z2 = np.copy(V_z)

print(n)
print(r)
print(num_samples)

#AltMin Steps
for i in range(num_iter):
        print(i)
        if(which_alg == "Matrix Alt Min" or which_alg == "all"):
                print("Matrix Alt Min")
                V_xmat, V_yzmat = matrix_altmin(V_xmat, V_yzmat)
                curr_error = eval_error_matrix(V_xmat, V_yzmat)
                print(curr_error)
                error.append(curr_error)

        if(which_alg == "Tensor Powering" or which_alg == "all"):
                print("Tensor Powering")
                if(curr_error > threshold):
                        V_x = power_altmin(V_x,V_y,V_z, x_dict)
                        V_y = power_altmin(V_y,V_z,V_x, y_dict)
                        V_z = power_altmin(V_z,V_x,V_y, z_dict)
                        curr_error = eval_error_direct(V_x,V_y,V_z,
                            x_dict)
                print(curr_error)
                error.append(curr_error)

        if(which_alg == "Subspace Powering" or which_alg == "all"):
                print("Subspace Powering")
                if(curr_error > threshold):
                        V_x2 = subspace_altmin(V_x2,V_y2,V_z2, x_dict)
                        V_y2 = subspace_altmin(V_y2,V_z2,V_x2, y_dict)
                        V_z2 = subspace_altmin(V_z2,V_x2,V_y2, z_dict)
                        curr_error = eval_error_subspace(V_x2,V_y2,
                            V_z2, x_dict)
                print(curr_error)
                error.append(curr_error)
```

```
all_errors.append(error)
to_save = np.transpose(np.array(all_errors))
avg_errors = np.mean(to_save, axis = 0)
np.savetxt(save_file, to_save, delimiter=",")
```