[Reviews · NeurIPS 2020]

Review 1

Summary and Contributions: This paper considers the tensor completion problem for a third order tensor. The CP decomposition, which decomposes a tensor into the sum of rank-1 tensor, is used. The alternative direction method with a twist is used to solve the problem. Theoretically, it is shown that the method (1) achieves exact recovery w.h.p.; (2) runs nearly linear time in terms of the number of observations; (3) converges at a linear rate; (4) scales to dimension.

Strengths: The "twist" in the alternative direction method is novel. Theoretically, it is shown that the method enjoys good properties -- (1)-(4) as above.

Weaknesses: For my understanding, the "twist" in some sense uses the Tucker decomposition, then the proposed method approximated it by a CP decompositionby solving a LS problem for H_{t+1}. It is unclear from the paper, how the LS problem is solved. Does the method scale with the dimension of the tensor and the number of observation? In Alg 2 and Thm 3.2, how to choose the parameters in practice is unclear. In the numerical experiments, the method is only compared with the standard alternative minimization method, other tensor completion methods should also be considered, e.g., Tmac. The response to my concerns is somewhat satisfactory. I will keep my score.

Correctness: The result seems reasonable, but I didn't check the proof.

Clarity: Yes.

Relation to Prior Work: A "twist" incorporated into the alternative minimization method for tensor completion, which is novel. The novel method enjoys good properties -- (1)-(4) as above.

Reproducibility: Yes

Additional Feedback:


Review 2

Summary and Contributions: This paper considers the tensor completion problem and proposes an efficient algorithm based on alternating minimization. The algorithm achieves exact completion in nearly linear time, and scales to large dimensions.

Strengths: The paper provides sound theoretical results as well as comprehensive numerical experiments. The proposed method works in the case where tensor factors are highly correlated, which is an important improvement over existing works. And the idea in this paper may find interesting for other problems in machine learning.

Weaknesses: The theory in Theorem 3.2 may seem loose in the dependence on incoherence parameter, rank and condition number, e.g. 300 in the power. I understand that authors are primarily interested in how the result scales with n. However, given that authors claim to make tensor completion practical, it might be better if the dependence on other parameters can be improved.

Correctness: I did not find major errors, and I believe them to be correct.

Clarity: This paper is well written.

Relation to Prior Work: This paper provides a nice review of and comparison with existing guarantees.

Reproducibility: Yes

Additional Feedback:


Review 3

Summary and Contributions: This paper proposes a new algorithm for tensor completion that offers both strong theoretical guarantees and a practical improvement: a rare combination! The main insight is that even when two subspaces are close, the Khatri-Rao product of their bases may not be --- but the Kronecker product is. The resulting idea is to run alternating minimization by choosing each factor to minimize a least-squares loss that involves a basis for the Kronecker product of the other two factors; standard AM for tensor completion uses the Khatri-Rao product. The resulting factor is too large (r^2 instead of r); the authors suggest reducing the dimension to r at each iteration with a standard low rank (matrix) approximation.

Strengths: By augmenting the method with a spectral initialization, a convex postprocessing procedure, sample splitting, and appropriate assumptions and parameters, the authors prove a theoretical guarantee that improves on the best available, showing exact recovery when O(n^{3/2}) samples are observed, and runs in time linear in the number of samples. Moreover, a few simple numerical experiments provide confidence that the simplest version of the method, Kronecker AM (with sample splitting), outperforms standard AM. Interestingly, the authors convincingly demonstrate that sample splitting improves the performance of standard AM, confounding this reviewer's previous intuition. (But I believe it.)

Weaknesses: The authors omit at least one important reference: Jain, Prateek, and Sewoong Oh. “Provable tensor factorization with missing data.” Advances in Neural Information Processing Systems. 2014. This paper also uses AM to complete a low-CP-rank tensor with the same sample complexity as the present paper (in n, up to polylogarithmic factors). The authors should also compare to this method in experiments, or justify why it’s not feasible to compare with. Nevertheless, the method presented here is so much simpler that I think it is a definite improvement.

Correctness: Yes, seems correct. I looked over the supplement but did not read the supplement in detail.

Clarity: Beautifully written; except for the figures! The plots are unacceptably formatted. The text all needs to be larger, preferably as large as the main text. It's impossible to read as is. Also, I recommend using different markers as well as colors to distinguish different series, for the benefit of colorblind readers or readers who print in b&w.

Relation to Prior Work: Mostly good; one notable exception is the Jain and Oh paper (see above).

Reproducibility: Yes

Additional Feedback: * Why do you suppose sample splitting improves performance of standard AM? * Did you try the method with spectral initialization and/or post-processing? Do they improve or worsen performance? * I'd suggest mentioning earlier in the paper that the method empirically works for noisy tensor completion, although you don't have theory yet. (This is not true of all algorithms, but super important from a practical perspective!) * The "parameters" (esp, as used in Theorem 3.2) of your algorithm are not parameters; they are data-generating assumptions (especially p). These values describe the given observations of the tensor, not the algorithmic choices we make in reconstructing it. * In algorithm 2, it seems the three alternating minimization problems are solved in parallel, not sequentially (eg, computing V^{t+1}_y from V^{t+1}_x and V^{t}_z). Any particular reason why? Do you think it would work either way? (In practice / in theory.) * Typo (a bad one!) in algorithm 2: What is U_x? * Typo in algorithm 2: paramter * Check your bibliography. Lots of capitalization errors.


Review 4

Summary and Contributions: The authors propose a different way of performing alternating optimization for low-rank tensor completion. Their idea is to use the Kronecker product, X \kron Y, instead of the Khatri-Rao product X \krp Y when they update the third matrix Z. This expands the number of dimensions from r to r^2, and they propose reducing that back to r by computing a principal subspace. They motivate this choice by arguing that the Kronecker product is more amenable to bounding the principal angle, which is convenient for their analysis. They provide a multi-step algorithm, several theoretical claims, and some illustrative experimental results.

Strengths: The tensor completion problem is very important to this and other communities.

Weaknesses: Modeling and motivation: In tensor decomposition, there are two basic models. One is tensor rank decomposition, equation (1) in this paper, where each x_i vector interacts with only one y_i and z_i (i.e., participates in only one tensor product) and the other is the Tucker model where every vector x_i potentially interacts with every other vector y_j and z_k - equation (2) in the paper. Without realizing it, the authors are actually mixing the two. What they end up doing in solving the Kronecker-structured subproblem is the ALS update for the Tucker decomposition model. After that, reducing back to the r principal vectors is reminiscent of orthonormal Tucker ALS, but doing it in two steps as the authors do sacrifices the monotone improvement property of ALS. Furthermore, the authors implicitly assume that the core is fixed and diagonal (because what they really want is r components). It is hard to understand the rationale of this approach. Assumptions: The authors never spell it out exactly, but they assume that tensor rank r is under n, whereas typical tensor rank is much higher than that and maximal rank is order of n^2. The methods considered simply cannot work for tensor rank r > n, which is in fact the typical case. The other implicit assumption is that the tensor can be decomposed using r orthogonal factors, which is not always true even for r < n, and the whitening transformation is very sensitive to noise. State of art and literature review: There are numerous tensor decomposition algorithms that can be used for completion, well beyond ALS. There are some widely used toolboxes, such as tensorlab, but none of these are included in the comparison. The authors only use "traditional ALS" that is randomly initialized, but it is well known that in the case of highly correlated factors other algorithms, such as G-N and L-M using trust region methods perform much better than randomly initialized ALS. Even with missing entries, clever initialization is possible. A lot of this know-how is already implemented in publicly available toolboxes, such as tensorlab, but the authors claim "Unfortunately many algorithms in the literature either cannot be run on real data, ... and for others no existing code was available." The authors should do a search on tensor decomposition software.

Correctness: Methods do not appear to be sound - see my earlier comments. Empirical methodology is definitely incorrect. The authors use toy examples, grossly inadequate baselines. See comments above about properly comparing with the state of the art in tensor decomposition and completion algorithms.

Clarity: Not really. There are many strong claims that are made, but are not substantiated. For example, the authors claim that their algorithm "scales to thousands of dimensions, whereas previous experiments had been limited to about a hundred dimensions", but that is not supported by experimental results. "Experimentally, in the presence of noise, it still achieves strong guarantees. In particular, it achieves nearly optimal prediction error." Optimal in what sense? Is there any statistical analysis to support this claim?

Relation to Prior Work: Not really. In fact the paper is missing numerous important references on tensor decomposition and completion algorithms that are far more principled and practical than the one proposed here.

Reproducibility: No

Additional Feedback: Given my concerns about the modeling and the author's lack of context regarding tensor decomposition/completion theory and algorithms (the cited literature leaves much to be desired), and the lack of clarity in the exposition, I didn't go through the 30 pages of supplementary material. Reviewer's additional comments after reading author's response: I'm not sure that the authors understand (they certainly don't want to admit) that, for tensors, r can be > n and still be (very) low rank. When maximal rank is of order n^2, r = 2 n is in fact very low. This case is not "bizarre" at all; the author's response is. All the results on optimal sample complexity that the authors refer to make the implicit assumption that r < n, otherwise there are of course more unknowns than equations. On practical tensor completion algorithms and the kinds of tensor sizes that they can handle, may I refer the authors to, e.g., https://doi.org/10.1016/j.parco.2017.11.002 which can handle datasets such as Amazon of size 4.8M X 1.7M X 1.8M of 1.4B nonzero entries and 41.5 GB? On the authors' numerical simulation example given in the rebuttal: Your experiment is simply not reproducible from the information provided (how were the rank-one factors drawn? Is this for a single tensor (as it appears to be)?What initialization was used for tensorlab? What were the stopping criteria used? Which of the many algorithms in tensorlab was used? With what parameters?). I would also really like to see how well their method works when the tensor is noisy - the case of a perfectly low-rank tensor of n=200 and r=4 is as far from practical as it gets. No real-world tensor is exactly low rank, even though most can be approximated as being low rank. But then the question is which method works best in this imperfect practice. Perhaps most importantly, the authors could have provided their code for the reviewers to check out during the review process - this is highly encouraged for Neurips submissions for reproducibility - but the authors chose not to do that. This is especially important to corroborate the theoretical claims, especially given the title of the paper: "Tensor completion made practical".


Review 5

Summary and Contributions: This paper proposes a new variant of the classical alternating least squares (ALS) algorithm to tackle tensor completion tasks under the CP low rank model. The main advantages of the proposed optimization algorithm is two-fold: it comes with strong theoretical guarantees and it is computationally efficient (can scale to very large tensors). The algorithm is an alternating minimization scheme where the classical step of ALS (minimizing wrt to one of the factor matrices while the other ones are keeping fixed) is replaced with a two step procedure (for a 3rd order target tensor): (i) find a rank R^2 approximation of the target tensor (where R is the target rank of the CP decomposition), where the terms of the fixed components corresponds to the pairwise Kronecker products of the columns of the fixed factor matrices. This is done by solving a least square problem wrt to the R^2 vectors of the mode being optimized. (ii) find the best rank R approximation of the matrix found in step (i) and use it as the new estimate for the factor matrix for this mode. Step (i) is crucial to the theoretical analysis as it allows the authors to control the principal angles between factor matrices at each iteration of the alternating minimization. This fundamental modification of the ALS algorithm is combined with a robust spectral based initialization based on the technique introduced by Montanari and Sun. Theoretical guarantees are given for the proposed algorithm which can be run in nearly linear time. Synthetic experiments are presented to showcase the benefits of the proposed approach where it is compared with the classical ALS algorithm and a variant of the proposed algorithm.

Strengths: - Except for a few minor typos the paper is very well written and easy to follow. - The theoretical guarantees are strong, building upon recent analysis of ALS guarantees for matrix completion. - The algorithm is computationally efficient and intuitive.

Weaknesses: - It is not clear how the method would scale to higher order tensors. It is true that for order 3 tensor the method is efficient, but from what I understand applying the proposed approach to e.g. an order 10 tensor would require solving a least square problem and a low rank approximation of size R^9 x n at each iteration. - The method seems to be limited to the undercomplete regime (R \leq n) and it is unclear whether it could be extended to the over-complete case. A discussion on this point is needed in my opinion. - The experimental section is a bit limited. The authors mention that some methods are not compared too as they cannot scale to large tensors but I think it would be possible and relevant to present experiments on smaller instances to compare with the SDP based methods. In summary, this paper makes a significant contribution as it presents a scalable algorithm for tensor completion coming with strong theoretical guarantees; furthermore, the proposed algorithm is a conceptually simple modification of the classical ALS algorithm, which is very commonly used for tensor problems. On the theoretical side, I believe this contribution is very significant and relevant to the NeurIPS community. On the other side, it is true that the experimental evaluation is a bit lacking, but it sill shows that the proposed approach performs significantly better than ALS (with not much increase in computational cost). Given that ALS is very commonly used in practice and the conceptual simplicity of the proposed algorithm I believe that it has the potential of being adopted by practitioners. More importantly regarding the weaknesses, I feel that the limitations of the proposed approach are not discussed thoroughly enough in the paper.

Correctness: I did not check the proofs so I cannot assess the correctness of the theoretical result but they seem sound. The empirical methodology is correct.

Clarity: Yes the paper is overall well written though it would benefit from additional details and discussions at some places.

Relation to Prior Work: Yes the relation to prior work is clearly discussed but more alternative approaches could be included in the experimental comparison.

Reproducibility: Yes

Additional Feedback: - the quality of the Figures in the experiment section is quite poor. - line 17, typo: u_i v_i w_i should x_i y_i z_i - the notation T|_S has not been introduced and I don't think it is a standard notation (at least I am not familiar with it). - line 159, typo: will *be* close to T

[Author Response · NeurIPS 2020]

**Author Feedback:** Mostly, we would like to address Reviewer #4 who has misunderstood virtually all parts of our paper. First, Reviewer #4 claims that our paper only works for orthogonal tensors. This is not true and is actually the main point of the paper. As we discussed at length in Section 1.1 the only previous works *with provable guarantees* needed to assume orthogonality, near orthogonality or used a very large semidefinite program. We use PCA in each iteration, but as we explained, the goal is to find an approximation to the subspace spanned by the unknown factors (whether they be orthonormal or not) and then in a postprocessing step that works via Jennrich's algorithm to find factors that span this space that work (but again, they need not be orthonormal).

Second, Reviewer #4 has seemingly missed the entire point of the paper that the goal is to provide the first theoretical guarantees for exact tensor completion (and as a bonus we get a practical new algorithm). Instead Reviewer #4 opines that we sacrifice the monotone improvement property of ALS. So? ALS has no known provable guarantees and as we show experimentally can get stuck in suboptimal local minima. Instead we give a new method that provably works and achieves linear convergence rate and never gets stuck. The goal of our paper is to discover new ways to do things. These involve subtle changes and Reviewer #4 has seemingly misunderstood how our algorithm works and instead attempts to draw faulty analogies with other existing methods.

Third, Reviewer #4 makes some bizarre claims, such as arguing that the $r > n$ case is the most interesting and natural setting. Tensor decompositions and completions are used in practice in many applications tensors are at least close to being low rank. In fact a google search for the exact phrase "Low Rank Tensor Completion" yields over 35k results.

That said Reviewer #4 brings up a fair point that we could have included more experimental comparisons, e.g. with other heuristics that are out there that do not have provable guarantees. First, as has been documented many times in the literature (see [1]), standard baselines (ADMM, FaLRTC, TMac, TTN) that require storing the whole tensor in memory run out of memory even on tensors that have dimensions $15000 \times 15000 \times 5$. These tensors have about as many entries as the tensors in our largest experiments. Second we tried out the CPD method in tensorlab which indeed performs much worse than our method. For example, when we have a random tensor with $n = 200, r = 4$ and the number of observations is 50000 our method reliably achieves very small $10^{-6}$ error whereas CPD reliably achieves error rates that are orders of magnitude worse, between $10^{-1}$ and $10^0$. Only when the number of observations is much larger (200000) does CPD start to be competitive.

|  | Kronecker Comp. (50k) | Tensorlab CPD (50k) | Kronecker Comp. (200k) | Tensorlab CPD (200k) |
|---|---|---|---|---|
| $10^{-1} < \text{err} < 10^0$ | 1 | 36 | 0 | 10 |
| $10^{-2} < \text{err} < 10^{-1}$ | 0 | 4 | 0 | 5 |
| $10^{-6} < \text{err} < 10^{-2}$ | 1 | 0 | 0 | 2 |
| $\text{err} < 10^{-6}$ | 38 | 0 | 40 | 23 |

Figure 1: Entries are the number of trials out of 40 that achieved a certain relative RMSE. Kronecker Comp. (our algorithm) is run for 100 iterations. Tensorlab CPD is run with all default settings.

Reviewer #3 asks about the relation between our work and the paper "Provable tensor factorization with missing data". In fact, they assume that the factors are orthogonal. In contrast, ours is the first algorithm to achieve exact recovery while allowing the factors to be strongly correlated.

Reviewer #1 asks about the complexity of the LS step. The LS problem we are solving involves $r^{O(1)}$ variables and so it can be straightforwardly solved in nearly linear time in the number of observations (when $r$ is polylogarithmic) using Gram-Schmidt. We will clarify this in our paper. The only important point is that, while the vectors have dimension $n^2$, they are sparse so we just need to store their nonzero entries as lists.

Reviewer #1 also asks about how to set the parameters in practice. We can always terminate the iterative step early if we reach small enough error. The important thing is how the parameters in Theorem 3.2 should be set, and we find that in practice changing the 300 in the exponent to 2 suffices. Our focus was the dependence of our sample complexity and running time on $n$, but seemingly it does quite well in terms of $r$ too, which is necessary for the kinds of experimental results we get.

[1] Xiawei Guo, Quanming Yao, and James Tin-Yau Kwok. Efficient sparse low-rank tensor completion using the frank-wolfe algorithm. In *Thirty-First AAAI Conference on Artificial Intelligence*, 2017.



[Meta-Review · NeurIPS 2020]

This paper proposes a "twist" in the usual alternating direction method for CP, and proves results that some reviewers found surprising (in a good way). As R3, said, it offers a rare combination of theoretical guarantees and practical improvement. R4 brought up some issues, most notably with the assumption of the tensor rank not being too large, as well as concerns about the experiments and not providing code. Several reviewers wanted to see comparisons with more methods (beyond ALS). There was also an overall concern that with 30 pages of supplement, it was unrealistic for reviewers to be expected to check all details of all proofs. In the AC's own view, these kinds of submissions are not appropriate for NeurIPS, where reviewers have very limited time and many papers to review. Due to the range of reviewer disagreements, we added an additional reviewer, who generally found the paper above-the-bar, but was worried the authors would completely ignore some of R4's comments. In particular, R5 wants (1) discussion about limitation to under-complete case and scaling to higher order, (2) comparisons beyond ALS, (3) add literature as pointed out by R4 and R3, and (4) fix tables as pointed out by R3. Overall, we feel that this is a paper that could potentially have a large impact, and therefore recommend accept.